# Risk-Sensitive Theory of Mind: Coordinating with Agents of Unknown Bias using Cumulative Prospect Theory

**Mason O. Smith** [1]   **Wenlong Zhang** [1]

## Abstract

Humans are often modeled as rational actors by interactive agents when they are in fact frequently observed to make biased decisions. This erroneous assumption may cause an agent's model of the human to fail, especially when interaction occurs in bias-inducing settings that prompt risky decisions. To address this, this paper formulates a risk-sensitive multi-agent coordination problem and presents the novel Risk-Sensitive Theory of Mind (RS-ToM) framework that allows an autonomous agent to reason about and adapt to a partner of unknown risk-sensitivity. In simulated studies, we show that an agent with an RS-ToM is able to better coordinate with such a partner when compared to an agent that assumes their partner is rational. Thus, we observe significant improvements to team performance, coordination fluency, compliance with partner risk-preferences, and predictability. The presented results suggest that an RS-ToM will be able to model and plan with partners that exhibit these risk-sensitive biases in the real world.

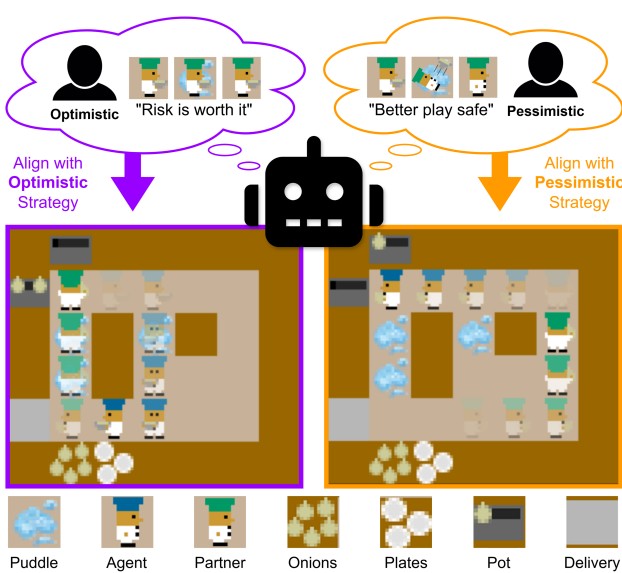

*Figure 1.* Illustration of a (robot) agent reasoning about how its (human) partner may hold irrationally optimistic or pessimistic perceptions about entering a puddle, where they risk slipping and losing their held item, and the best way to align its strategy to improve team coordination in the Risky Overcooked task.

## 1. Introduction

Autonomous agents have traditionally modeled their human partners as noisy-rational actors (Baker & Tenenbaum, 2014) and think in approximately the same way that machines do. While convenient, this can be an erroneous assumption as humans often display behavior that systemically deviates from rational strategies. This is referred to as biased decision-making (Kahneman & Tversky, 1974). Most relevant to the proposed work, humans tend to be either irrationally optimistic (i.e., risk-seeking) or pessimistic (i.e.,

risk-averse) about their prospects when faced with a risky decision (Kahneman & Tversky, 2013). For example, consider the task in Fig. 1 which demonstrates two strategies. An optimistic human would likely prefer the purple strategy that risks traversing puddle states since optimism discounts the negative consequence of slipping. Conversely, a pessimistic human would likely prefer the orange strategy and take an unnecessarily long detour to avoid any risk at all. An autonomous agent that can reason about these preferences can then deploy strategies that is best aligned with (biased) human values to improve coordination.

Humans are said to have *risk-sensitive* preferences that violate axioms of rational decision-making (Briggs, 2023) that noisy-rationality relies upon. When violated, this leads to the inability for an agent to understand human preferences and accurately predict their actions. Thus, team performance can suffer when inadequate consideration of human

[1]School of Manufacturing Systems and Networks, Arizona State University, Mesa, Arizona USA. Correspondence to: Mason O. Smith <mosmith3@asu.edu>.

*Proceedings of the 42nd International Conference on Machine Learning*, Vancouver, Canada. PMLR 267, 2025. Copyright 2025 by the author(s).

risk-sensitivity is given in settings that prompt such biases to occur (Kwon et al., 2020). Consideration of risk-sensitivity may also impact interpersonal dynamics like trust (Smith & Zhang, 2025), a critical feature of human-autonomy interaction (HAT). In fact, risk-sensitivity was chosen as the target for improving an agent's model of their partner due to its unique connection to trust: trust requires both *risk* and truster (human) *vulnerability* to be present (Lee & Moray, 1992). Therefore, every trusting situation can prompt risk-sensitive biases to occur. Automation that is then able to infer and reason about this relationship may then be afforded a more nuanced approach to closing the loop with trust calibration in HAT.

Motivated by these opportunities from improving upon the assumption of noisy-rationality, this paper formulates a risk-sensitive multi-agent coordination problem where an autonomous agent must coordinate with a partner of unknown risk-sensitivity. We then attempt to solve this problem by reasoning about the partner's biased preferences as different than our own, referred to as having a Theory of Mind (ToM) (Devin & Alami, 2016), and having the agent align with its partner's preferences. An ToM classically reasons over a discrete set of intentions or subtasks (Wu et al., 2021) as a compact way to define preferences within the context of a single task. We take a more abstract perspective that can ubiquitously model preferences in any risky task on a continuous spectrum by varying model parameters. Therefore, when specifically reasoning about the partner's risk-sensitive preferences, we refer to this as an Risk-Sensitive Theory of Mind (RS-ToM). Here, we seek to evaluate 1) how to overcome learning challenges of risk-sensitive coordination problems in complex, multi-agent settings, 2) what, if any, improvements to team coordination are present in contrast to the aforementioned assumption of noisy-rationality, and 3) what are the implications of the simulated results on HAT in terms of team coordination and trust?

**Related Works.** An RS-ToM draws inspiration from risk-aware control settings that bound an agent's future exposure to risk by applying pessimistic risk measures like conditional value-at-risk (CVaR) (Hakobyan et al., 2019; Tan et al., 2022). More closely related to the proposed work is the subset of approaches that apply more generalized risk measures like Cumulative Prospect Theory (CPT) that better align with empirical observations of human risk-sensitivity (Tversky & Kahneman, 1992). Works that apply CPT in sequential decision tasks tend to impart prototypical risk-sensitivity onto an agent such that it behaves more "human-like" and its actions can be better understood by humans (Ramasubramanian et al., 2021; Prashanth et al., 2016). Closest in formulation to an RS-ToM, the authors of (Danis et al., 2023) applied CPT in a multi-agent setting using Nash Q-Learning to generate risk-sensitive joint strategies in a simple grid-world navigation task.

While these approaches are a good step towards better aligning autonomy and biased human values, they all train policies under the median CPT parameter estimates from (Tversky & Kahneman, 1992) which generally describe risk-averse behaviors. Thus, in practice, they align with a single type of risk-sensitivity prescribed to the human and emulate a single, fixed mode of bias. This fails to capture a powerful aspect of CPT that distinguishes it from other measures like CVaR: CPT can model a variety of risk-sensitive modes including risk-seeking tendencies. This may be important to consider as human risk-sensitivity can vary between individuals and contexts (Kahneman & Tversky, 2013) and learned coordination strategies can drastically change based on how CPT is parameterized (Ferreira et al., 2021). Thus, an agent afforded the ability to reason over several modes of risk-sensitive behaviors (i.e. has an RS-ToM) can adapt online to different contexts and personalize on an individual level during interaction.

Works on personalized risk-sensitive models are limited. In (Kwon et al., 2020), the authors collected a data-prior to formulate a personalized model of human risk-sensitivity with CPT in a collaborative cup stacking task. Notably, they showed that a robot endowed with a personalized model can improve team performance and perceived trustworthiness of the robot by better aligning with risk-sensitive preferences. Similarly, (Sun et al., 2019) learned CPT-based utility functions using inverse reinforcement learning (IRL) in a roundabout driving task. Constraining IRL with learned CPT parameters unlocks personalized explanations of irrational behaviors not available under noisy-rationality that lead to improved human-prediction accuracy and coordination. Alternatively, the authors in (Cheng et al., 2023) used interactive querying to estimate CPT parameters. The main limitation that these works share is thr reliance on a human data-prior which may be expensive or infeasible to collect, especially when it requires human risk to be present. Additionally, these approaches can only account for differences between humans, not how each's risk-sensitivity changes between contexts or over time.

Lastly, works on value-based multi-agent reinforcement learning (MARL) with CPT tend to implement tabular algorithms (e.g., value iteration (Ferreira et al., 2021; Kwon et al., 2020), Q-learning (Danis et al., 2023), or SARSA (Ramasubramanian et al., 2021)) to solve low-dimensional learning problems. This limits task complexity and the ability to learn in real-world settings. Thus, there is a gap in existing value-based methods that are able to solve these problems as deep learning approaches are not well studied in this space.

**Our Approach.**

As prior works have shown that adaptation to human risk-preferences can benefit interaction, this paper focuses on

*algorithmic contributions* that address the aforementioned challenges (i.e., data prior requirements to enable personalization and the limited task complexity afforded with tabular methods) where we seek to validate with real humans in future studies. To the best of the authors' knowledge, we are the first to formulate an RS-ToM that affords adaptation to a partner with unknown risk-sensitivity in a zero-shot fashion and without the need for costly data-priors. We achieve this by first drawing from existing work in risk-sensitive reinforcement learning (Danis et al., 2023) to train several candidate policies conditioned on a space of CPT parameters. This is a similar goal to using IRL to recover suboptimal value functions (Rothkopf & Dimitrakakis, 2011; Bergerson, 2021) where we instead take the forward approach that avoids data requirements by prescribing agents a CPT-based evaluation of task objectives. Then, when interacting with a partner of unknown risk-sensitivity, an agent will infer which candidate policy best describes the current partner's behavior and deploy the appropriate policy in response. Doing so, we enable the autonomous agent to align its preferences with different types of risk-sensitivity so that it can better coordinate. To that end, the contributions of this paper are as follows:

1. To our knowledge, we are the first to integrate CPT into deep multi-agent risk-sensitive reinforcement learning (MARSRL) to induce biased behavior in complex risky-decision tasks.

2. We formulate a novel memory buffer and apply a level-k quantal response equilibrium to overcome tractability issues in this deep multi-agent setting with CPT.

3. We then leverage our MARSRL algorithm to enable an RS-ToM and demonstrate its ability to achieve superior team performance, coordination fluency, risk taking alignment, and partner predictability relative to a noisy-rational baseline when interacting with a partner of unknown risk-sensitivity in our novel Risky Overcooked benchmark task.

The remainder of this paper is as follows. Section 2 provides a formulation of the RS-ToM. Section 3 details the experimental procedure including description of the Risky Overcooked task (Sec. 3.1), analysis methods (Sec. 3.2), and results (Sec. 3.3). Results and implications are discussed in Sec. 4 and concluding remarks are given in Sec. 5.

## 2. Risk-Sensitive Theory of Mind

An RS-ToM is derived of three components. First, we will define CPT as the risk-measure that induces risk-sensitive evaluation of prospects. Next, we will define the approach to the MARL problem and how it is extended to MARSRL using CPT where we leverage knowledge of the transition

model and a novel replay memory for improved tractability. The last component is what affords an agent with an RS-ToM and will describe inference over and response to a space of pre-trained risk-sensitive policies. The remainder section will formulate these three steps while a high-level summary of this framework can be found in Fig. 2.

### 2.1. Cumulative Prospect Theory

Let $\mathcal{X}$ denote the support of a discrete random variable with probability $p_i$ of having utility $\mathcal{X}_i$ s.t. $\{\mathcal{X}_i, p_i\} \in \mathcal{X}$. Following (Tversky & Kahneman, 1992), we then define the transformations on utility and probability.

**Definition 2.1.** *Let $u^+(\cdot)$ and $u^-(\cdot)$ denote the utility transformations applied to what the agent perceives as gains and losses, respectively, and be defined as:*

$$u^+(\mathcal{X}_i) = (\mathcal{X}_i - b)^{\eta^+}$$
$$u^-(\mathcal{X}_i) = \ell|\mathcal{X}_i - b|^{\eta^-} \tag{1}$$

where $\eta^+, \eta^- \in [0, 1]$ account for exponential discounting of utility as it moves farther from the reference point $b \in (-\infty, \infty)$ and $\ell \in [0, \infty)$ asymmetrically weights losses.

**Definition 2.2.** *Let $w^+(\cdot)$ and $w^-(\cdot)$ denote the probability transformations applied to what the agent perceives as gains and losses, respectively, and be defined as:*

$$w^+(p_i) = \frac{p_i^{\delta^+}}{(p_i^{\delta^+} + (1 - p_i)^{\delta^+})^{1/\delta^+}}$$
$$w^-(p_i) = \frac{p_i^{\delta^-}}{(p_i^{\delta^-} + (1 - p_i)^{\delta^-})^{1/\delta^-}} \tag{2}$$

where $\delta^+, \delta^- \in [0, 1]$ apply inverted S-shaped transformations on probabilities that overestimates small probabilities and underestimates large probabilities. Examples of these transformations and more intuitive descriptions of each parameter can be found in Appendix B.1.

In order to deal with more than two prospects, expectations are carried out over the cumulative probability distributions.

**Definition 2.3.** *Let $\mathcal{X}$ be arranged in increasing order of $\mathcal{X}_i$ s.t. $\mathcal{X}_1 \leq \mathcal{X}_2 \leq ... \leq \mathcal{X}_K$ and $l$ be the index where $\mathcal{X}_i \leq b$. Also, define $F_k := \sum_{i=1}^{k} p_i \ \forall \ k \leq l$ and $\sum_{i=k}^{K} p_i \ \forall \ k > l$ as the cumulative probability distributions for gains and losses, respectively. We then define CPT-expectation of $\mathcal{X}$ as $\rho_{cpt}(\mathcal{X})$:*

$$\rho_{cpt}(\mathcal{X}) =$$
$$\left( \sum_{i=l+1}^{K-1} u^+(\mathcal{X}_i)\left(w^+(F_i) - w^+(F_{i+1})\right) + u^+(\mathcal{X}_K)w^+(p_K) \right)$$
$$- \left( u^-(\mathcal{X}_1)w^-(p_1) + \sum_{i=2}^{l} u^-(\mathcal{X}_i)(w^-(F_i) - w^-(F_{i-1})) \right) \tag{3}$$

*Figure 2.* Illustration of the RS-ToM framework. The offline MARSRL algorithm trains a space of risk-sensitive candidate policies conditioned on the CPT-value transformation $\rho_{cpt}(\boldsymbol{\tau})$. The red lines show the novel replay memory that stores all possible next states $s'_i$ and their probability $p_i$. During the update of $Q_{cpt}$, the value $V_{cpt}(\hat{s}'_i)$ of each prospective state $\hat{s}'_i$ is calculated from the expectation over the level-k quantal response equilibrium (QRE) policy and the forward pass of the DDQN $Q_{cpt}(\hat{s}'_i)$. An agent of arbitrary risk-sensitivity can be generated by tuning the CPT parameters that defines the transformation in the red shaded region. Thus, any number of candidate policies can be trained such that an agent can reason about its partner's risk-sensitivity online, in a zero-shot fashion, and without the need for prior interaction data by performing a belief update over the space of pretrained candidates.

where $u^+, u^- : \mathbb{R} \to \mathbb{R}_+$ are continuous, have a bounded moments s.t. $u^+(\mathcal{X}_i) = 0 \ \forall \ \mathcal{X}_i \leq b$ and $u^-(\mathcal{X}_i) = 0 \ \forall \ \mathcal{X}_i > b$ , and are monotonically non-decreasing otherwise. Also, we assume the probability weighting functions $w^+, w^- : [0,1] \to [0,1]$ are Lipschitz continuous (Prashanth et al., 2016) and non-decreasing such that they satisfy $w^+(0), w^-(0) = 0$ and $w^+(1), w^-(1) = 1$.

## 2.2. MARSRL Algorithm

**Definition 2.4.** *An MDP is the tuple $\mathcal{M} := (S, A, \mathbb{P}, r)$ where $S : S_k \times S_{\text{-}k}$ is a finite set of joint-states for agent k and their partner -k, $A : A_k \times A_{\text{-}k}$ is the finite set of joint-actions that are sampled from the joint-policy $\pi : \pi_k \times \pi_{\text{-}k}$, $\mathbb{P}(s' \mid s, a) : S \times A \times S$ is the probability of transitioning to state $s'$ when action $a$ is taken in state $s$, and $r(s, a)$ is the stochastic reward function.*

The goal of standard reinforcement learning is then to find a policy $\pi$ that maximizes an objective $\sum_{t=0}^{T} \mathbb{E}[\gamma^t r(s, a) \mid \pi]$ over some time horizon $T$ where $\gamma \in (0, 1)$ is a discounting factor weighting more immediate rewards as more important (Sutton et al., 1998). We specify a policy by the quality of a state-action pair $Q(s, a)$ where $\pi(s, a) \propto exp(\lambda Q(s, a))$ and $\lambda$ defines the decision temperature of a policy. Given that the value of a state can be written as $V(s \mid \pi) = \sum_{a \in A} \pi(a \mid s) Q(s, a)$, we can define $Q(s, a)$ as:

$$Q(s, a) = r(s, a) + \gamma V(s' \mid \pi) \quad (4)$$

Given that states stochastically evolve according to $\mathbb{P}(s' \mid s, a)$, we can generalize (4) to the following expected value

over all possible next states $s'_i \in S$:

$$Q(s, a) = \mathbb{E}_{s'}[r(s, a) + \gamma V(s' \mid \pi)]$$
$$= r(s, a) + \gamma \sum_{s'_i \in S} \mathbb{P}(s'_i \mid s, a) V(s'_i \mid \pi) \quad (5)$$

We can then apply dynamic programming methods to iteratively update $Q(s, a)$ such that:

$$Q(s, a) \leftarrow Q(s, a) + \alpha \Big( \tau - Q(s, a) \Big) \quad (6)$$

where $\tau := r(s, a) + \gamma \sum_{s'_i \in S} \mathbb{P}(s'_i \mid s, a) V(s'_i \mid \pi)$ is refereed to as the temporal difference target (TD-Target) and $\alpha$ is the learning rate. In multi-agent settings, we recover the joint-policy $\pi(a \mid s) = (\pi_k, \pi_{\text{-}k})$, where $p(a \mid s) = \pi_k(a_k \mid s)\pi_{\text{-}k}(a_{\text{-}k} \mid s)$, by solving each transition as a single-stage, common-payoff game $Q(s)$ using the Quantal Response Equilibria (QRE) (see Appendix A.4 for details). This reinforcement learning algorithm and its following risk-sensitive version can be arbitrarily extended to $n > 2$ agents with appropriate factorization of the state-action space and equilibrium solution.

**CPT-Value.** The previous formulation can now be modified with CPT by treating the TD-target $\tau$ as the random variable $\mathcal{X}_i$ in (3) by defining:

$$\tau_i = r(s, a) + \gamma V(s'_i \mid \pi)$$
$$p_i = \mathbb{P}(s'_i \mid s, a) \quad (7)$$

where $p_i$ is the chance of observing $\tau_i$ as the TD-target and $i$ is the index of each possible next state $s'_i$ given action $a$ was taken in state $s$. While other works employ a sample-based approximation of CPT-value to maintain a model-free approach (Ramasubramanian et al., 2021; Danis et al., 2023;

Prashanth et al., 2016), we instead leverage knowledge of the environment for improved tractability. Thus, we directly compute the CPT-value $\rho_{cpt}(\boldsymbol{\tau})$ over the finite set of possible TD-targets $\boldsymbol{\tau}$ s.t. $\{\tau_i, p_i\} \in \boldsymbol{\tau}$ using (3) and rewrite the update from (6) in its risk-sensitive form:

$$Q_{cpt}(s,a) \leftarrow Q_{cpt}(s,a) + \alpha\Big(\rho_{cpt}(\boldsymbol{\tau}) - Q_{cpt}(s,a)\Big) \quad (8)$$

**Deep Learning:** While similar work has been conducted in simple, multi-agent settings using tabular methods (Danis et al., 2023; Ferreira et al., 2021) and more complex, single-agent settings using deep learning methods (Ramasubramanian et al., 2021; Rajabi et al., 2022), our work focuses on complex multi-agent tasks which calls for deep learning. We therefore solve (8), by applying Algorithm 1 which is based on a Double Deep Q-Network (DDQN) (van Hasselt et al., 2016) that keeps track of separate policy $Q_{cpt}^{\theta}$ and target networks $Q_{cpt}^{\phi}$ defined with weights on $\theta$ and $\phi$, respectively. Selected hyperparamters for learning can be found in Appendix A.3.

However, CPT induces additional computational challenges in a deep setting. In contrast to a rational expectation, CPT increases complexity by a factor of $3n+n^2+nlog(n)$ where $n$ is the number of prospects included in the expectation[1]. Also, CPT-value requires computation of all possible next states for each minibatch sample[2] which is only exacerbated by computation of equilibrium solutions. To overcome this, we first apply the QRE as an efficient approximation of a mixed Nash equilibrium solution. We then formulate a novel replay memory $\mathcal{D}$ that, instead of storing the traditional observed transition $(s, a, r, s')$, stores all transition prospects in the form of $(s, a, r, \mathbf{s}')$ where $\{s_i', p_i\} \in \mathbf{s}'$ is all possible next-states. Thus, given we update every timestep, we compute possible next states once instead of $N$ times.

To further aid tractability in the later Risky Overcooked task, we apply curriculum learning and reward shaping strategies (see Appendix A.2 for details) to deal with sparse rewards. Moreover, due to the symmetrical roles of the task, we leverage self-play methods (i.e. single Q-function for both agents) where a transition prospect is stored from each agent's perspective $\mathcal{D} \leftarrow (s, a, r, \mathbf{s}')_k \; \forall \; k$ at every timestep to improve sample efficiency.

We establish the convergence of Algorithm 1 by showing the update of $Q_{cpt}$ will be a contraction given appropriate choice of learning parameters. Also, we rely on a conservative upper-bound assumption on $\ell$ that ensures convergence of the Bellman operator (see Appendix A.5 for details).

---

**Algorithm 1** Deep MARSRL with CPT

**Input:** exploration rate $\epsilon < 1$, soft update rate $\alpha << 1$, and parametrized CPT-value functional $\rho_{cpt}(\cdot)$
**Initialize:** Policy network $Q_{cpt}^{\theta}$, target network $Q_{cpt}^{\phi}$, and replay memory $\mathcal{D} = \emptyset$
**for** episode $= 1, ..., M$ **do**
   Observe initial state $s$
   **while** $t \leq T$ **do**
      Sample action $a \sim \pi_{cpt}(s)$
      Observe next state $s'$ and reward $r$
      Get next state prospects $\mathbf{s}' := \{\{s_i', p_i\} : \forall s_i' \in S\}$
      Store transitions in memory $\mathcal{D} \leftarrow (s, a, r, \mathbf{s}')_k \; \forall k$
      Sample a $minibatch$ of $N$ transitions from $\mathcal{D}$
      **for all** $(\hat{s}, \hat{a}, \hat{r}, \hat{s}')_n \in minibatch$ **do**
        $\boldsymbol{\tau} = \{\{\hat{r} + \gamma V_{cpt}^{\phi}(\hat{s}_i'|\pi_{cpt}), p_i\} : \forall\{\hat{s}_i', p_i\} \in \hat{\mathbf{s}}'\}$
        $\mathcal{L}_n := (\rho_{cpt}(\boldsymbol{\tau}) - Q_{cpt}^{\theta}(\hat{s}, \hat{a}))^2$
      **end for**
      Perform gradient decent on $\theta$ with $\frac{1}{N}\sum_{n=1}^N \mathcal{L}_n$
      Soft target update $\phi \leftarrow \sigma\theta + (1-\sigma)\phi$
      $s \leftarrow s'$
   **end while**
**end for**

---

### 2.3. Inference

The inference step is what imparts the agent with an RS-ToM as it allows it to track its partner's risk-sensitive preferences independently from its own. Bayesian inference is performed over the past 10 observations $\mathcal{O} = \left\{\{s, a_{-k}\}_{t-10}, ..., \{s, a_{-k}\}_t\right\}$ to form a belief $b(\cdot)$ about which joint-policy $\pi \in \boldsymbol{\pi}$ the human $-k$ is following:

$$b(\boldsymbol{\pi} \mid \mathcal{O}) \leftarrow \frac{\mathbb{P}(\mathcal{O} \mid \boldsymbol{\pi})b(\boldsymbol{\pi})}{\sum\limits_{\pi \in \boldsymbol{\pi}} \mathbb{P}(\mathcal{O} \mid \pi)b(\pi)} \quad (9)$$

where $b(\boldsymbol{\pi} \mid \mathcal{O})$ is the updated belief over all possible joint-policies after observing $\mathcal{O}$. Here, we add a decaying weight to the likelihood $\mathbb{P}(\mathcal{O} \mid \boldsymbol{\pi})$ of observations further in the past[3]. The artificial agent $k$ then discretely adopts joint-strategy it believes is most likely s.t. $\pi_k \in \arg\max_{\pi} b(\pi|\mathcal{O})$. This is what frames the approach to coordination as a risk-sensitive alignment problem[4].

## 3. Experiments

The following experiments investigate RS-ToM's ability to adapt to a partner of unknown risk-sensitivity. Due to the novelty of an RS-ToM, there is a lack of feasible baseline algorithms that can adapt to partner risk-sensitivity in a zero-

---

[1]A rational expectation has linear complexity $n$ whereas sorting prospects has at best $nlog(n)$ complexity (Alkharabsheh et al., 2013), calculation of $F_k$ is $n^2$, and (1)-(3) all have $n$ complexity.

[2]This leads to $N-1$ more next state computations than tabular methods per Q-value update where $N$ is the minibatch size.

[3]This mirrors future human studies where recent observations are more representative of possibly dynamic risk-preferences.

[4]More sophisticated methods such as (Wang et al., 2025) are required for value alignment with $n > 2$ agents.

shot manner in complex settings. This stems from existing works for adapting to risk-sensitive preferences relying on data-priors collected from a partner (Kwon et al., 2020; Sun et al., 2019) and state of the art MARSRL algorithms (Danis et al., 2023; Ferreira et al., 2021) not being tractable in complex tasks like Risky Overcooked. However, we can evaluate performance in contrast with the standard assumption of noisy-rationality (RAT) to glean insight on expected outcomes when interacting with real humans.

### 3.1. Risky Overcooked

While there exists tasks such as traffic control (Prashanth et al., 2016) and multi-agent navigation (Danis et al., 2023; Tian et al., 2021), there appears to be no standardized risky decision benchmark that simultaneously incorporate shared awareness (mental models), a strong interdependence between teammates, optional reliance on one's partner, and explicit risk from the environment. This gap in standardized platforms is a common theme in HAT (Smith et al., 2024) that we seek to address by proposing Risky Overcooked. Risky Overcooked is an open-source, risky coordination benchmark that extends the popular Overcooked environment (Carroll et al., 2019) by incorporating risky decisions into team coordination.

In the standard Overcooked game, two agents must coordinate their behaviors to cook soup as fast as possible by completing three subtasks: 1) bring three *onions* to a *pot* at which time the soup will begin cooking, 2) bring a *plate* to the pot with a cooked soup to obtain *plated soup*, and 3) bring the plated soup to a *delivery window* to receive a reward of $r_{soup} = 20$. Agents cannot pass through each other, can only carry one object at a time, and can choose to place objects on any free counter space.

Overcooked task does not explicitly elicit risky decisions from the environment. Therefore, we propose the modification seen in Fig. 1 where we add risky puddle states. When an agent enters a risky state, they have a $p_\rho$ chance of slipping where they will lose whatever resource they are holding. The rationale of how values for $p_\rho$ are selected can be found in Appendix B.2. This frames the consequence of slipping as a forgone gain where we add a small time-cost $r_{tc} = -0.2$ at every timestep to elicit perceptions of loss if soups are not cooked fast enough. These risky states are intended to induce diverging and unique strategies for agents of different risk-sensitivities. Figure 3 shows how risk-averse teams tend to pessimistically avoid puddle states while risk-seeking teams optimistically traverse them (see Appendix B.3 for more examples).

Due to risk-sensitivity presenting itself as a choice between routes, we cannot rely on the mid-level action planners and state featurizations (Carroll et al., 2019) commonly used to simplify Overcooked. Instead, we are required to use low-level actions and a lossless state (see Appendix A.1 for details) that drastically increases complexity.

Design of Risky Overcooked layouts can be used to frame coordination in two ways where we select one of each framing for later experiments. First, the *risky/rely* framing proposes a choice between the level of acceptable risk and the reliance between partners. This is achieved by designing subtasks to require puddle state traversal or coordinate handoffs between agents. This framing couples risk-sensitivity closely with trust, as reliance is a behavioral outcome of trust (Lee & See, 2004), which makes it a promising setting for future human experiments. In the selected Risky Coordination Ring (RCR) layout, seen in the left half of Fig. 3, agents can either pass items through the center counter space to avoid all puddles (A and B in Fig. 3) or traverse puddles to complete subtasks independently (C and D in Fig. 3). Here, we see that a risk-averse strategy facilitates strong reliance between partners by always passing objects due to pessimistic perceptions of slipping while a risk-seeking strategy favors avoiding costs of imperfect handoffs due to optimistic perceptions.

Alternatively, the *risk/detour* framing proposes a choice between the level of acceptable risk and the additional navigation costs incurred by taking a detour around a puddle. In the selected Risky Multi-Path (RMP) layout, seen in the right half of Fig. 3, agents have three route choices: 1) a direct and high-risk route that traverses two puddles, 2) a slightly longer route that only requires traversal of one puddle, or 3) a long detour that requires no risk to be taken as it avoids all puddles. Here, we see that a risk-averse strategy tends to avoid puddle traversal by incurring higher navigation costs (E[5] and F in Fig. 3) while a risk-seeking strategy prefers to incur additional risk in favor of more direct routes (G and H in Fig. 3).

### 3.2. Analysis

A space of three joint-policy candidates $\pi$ containing risk-seeking $\pi_S$, risk-neutral (i.e. rational) $\pi_0$, and risk-averse $\pi_A$ strategies are trained with the MARSRL algorithm using CPT parameters motivated by median estimates from (Tversky & Kahneman, 1992) (see Appendix B.1 for further description). We then evaluate the difference between an agent with an RS-ToM and a RAT assumption when interacting with a partner of unknown risk-sensitivity. We also provide results contrasted against an Oracle baseline where the ground truth risk-sensitivity of the partner is known to show theoretically optimal results. The rest of this section will describe the four measures used to evaluate these experiments.

---

[5]Traversal of puddles is caused by the other agent blocking the risk-averse route when picking up a dish.

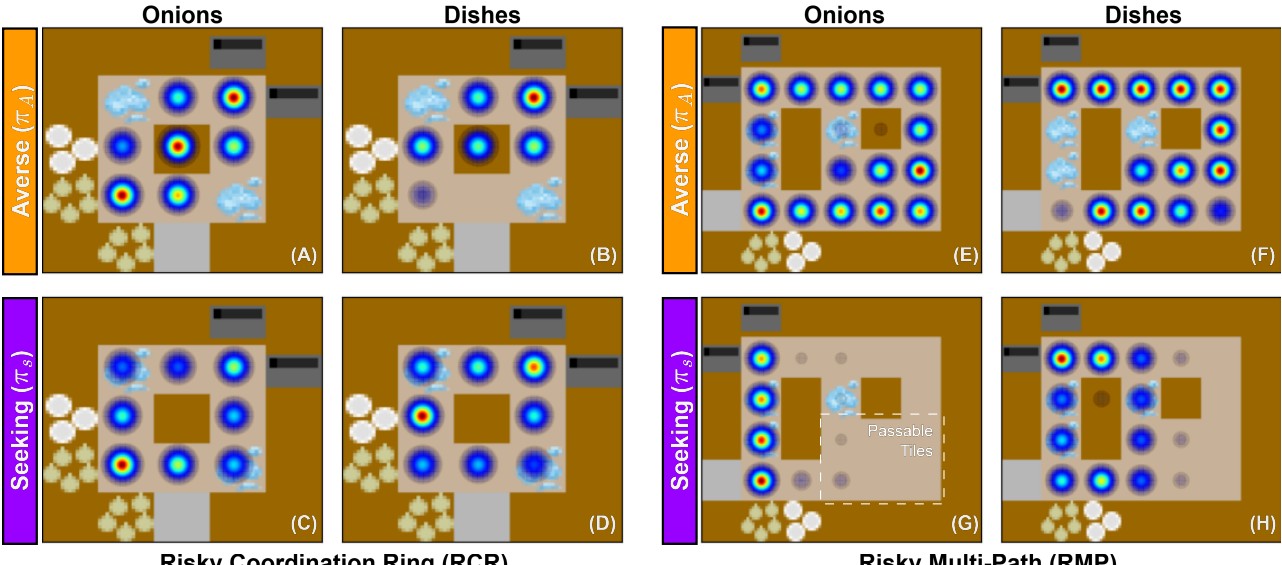

*Figure 3.* Illustration of successful generation of various joint strategies with the proposed MARSRL algorithm by showing the state visitation frequencies of the onion and dish objects carried by either agent where higher frequencies are indicated by increasingly red centroids in each tile. The top row shows the learned Averse $\pi_A$ strategy while the bottom row shows the Seeking $\pi_S$ strategy for both the RCR and RMP layouts. Soup visitation frequencies (excluded) always follows a path that avoids puddles due to the high cost of slipping being irrational, regardless of risk-sensitivity. The $2 \times 3$ area in (G) shows the space where agents may freely pass by each other.

Team performance in the Risky Overcooked task will be evaluated as the *cumulative reward of the team* $\Sigma r$. The time-cost penalty $r_{tc}$ is removed during calculation for simpler analysis where $\Sigma r$ is always positive.

Coordination fluency is the "coordinated meshing of joint activities of members of a well-synchronized team" and will be evaluated with *concurrent activity* C-ACT (Hoffman, 2019) defined as the proportion of task time where both agents are active simultaneously. A high C-ACT indicates a fair work balance and good synchronization of strategies. In Risky Overcooked, an agent is considered inactive while remaining in the same state or during periods that result in the return to a previous state without completion of a subtask[6].

The *number of risks taken by the team* $N_{risk}$ accounts for how well an agent aligns to their partner's risk-taking preferences. When interacting with a human, they would likely judge an agent's strategy as *inappropriate* if the amount of acceptable risk realized by the team is not properly calibrated to their perceptions. For example, a risk-averse human may view an agent taking the optimal amount of risk as "too risky" while a risk-seeking human may view it as "too safe."

---

[6]This accounts for an agent moving back and forth or having to retrace their steps in order to get out of the way of their partner in a tight corridor. In both cases, the behavior is not productive and is equivalent to an agent waiting in the same state.

The *predictability of the risk-sensitive partner* $\mathcal{P}_{\mathcal{H}}$ is observed to evaluate the afforded ability for the agent to predict future actions of the risk-sensitive partner and plan coordinated strategies. This is defined as the probability that the risk-sensitive partner takes the maximum likelihood action under the agent's (inferred) model of partner's risk-sensitivity. We are primarily interested in improving the predictability of the risk-sensitive partner and therefore do not evaluate their predictability of the other agent.

### 3.3. Results

The mean values of $N = 1000$ trials for the Oracle, RS-ToM, and RAT algorithms interacting with a Seeking or Averse partner across the RCR and RMP layouts can be found in Table 1. Bold values indicate the superior performance between RS-ToM and RAT.

A $2 \times 2$ ANOVA, with factors of 1) partner risk-sensitivity (Seeking and Averse) that is unknown to the ego agent and 2) the ego agent's algorithm (RS-ToM or RAT), was used to evaluate significance within each layout. Both main effects and all interaction effects were significant for all measures. A Tukey post-hoc test revealed all but five significant comparisons that will be mentioned when relevant. Unless stated otherwise, all tests had a significance level of $p < 0.02$. As ideal values of measures depends on the layout, results are normalized with percent change from RAT to RS-ToM for easier discussion.

*Table 1.* Numerical Results from Risky-Overcooked Experiments

| Layout | Partner | $\Sigma r$ | | | C-ACT | | | $N_{risk}$ | | | $\mathcal{P}_H$ | | |
|---|---|---|---|---|---|---|---|---|---|---|---|---|---|
| | | Oracle | RS-ToM | RAT | Oracle | RS-ToM | RAT | Oracle | RS-ToM | RAT | Oracle | RS-ToM | RAT |
| RCR | Seeking | 151 | **150** | 98 | 0.64 | **0.64** | 0.44 | 25.60 | 25.93 | 26.96 | 0.76 | **0.76** | 0.41 |
| | Averse | 146 | **146** | 70 | 0.48 | **0.47** | 0.26 | 0.00 | **0.02** | 8.42 | 0.75 | **0.75** | 0.26 |
| RMP | Seeking | 125 | **122** | 33 | 0.40 | **0.40** | 0.15 | 73.08 | 72.59 | 29.84 | 0.65 | **0.65** | 0.26 |
| | Averse | 100 | **101** | 44 | 0.51 | **0.51** | 0.22 | 14.79 | **15.07** | 35.24 | 0.65 | **0.65** | 0.33 |

**Comparison to Oracle.** In all experiments, the RS-ToM algorithm performed nearly identically to the Oracle. This is due to the RS-ToM's inference correctly converging to the ground truth quickly as seen by Fig 4. The RAT algorithm saw significant losses in all cases except for $N_{risk}$ in the RCR with a Seeking partner. Rationale for this is explained in the later "Risk Taking Alignment" results.

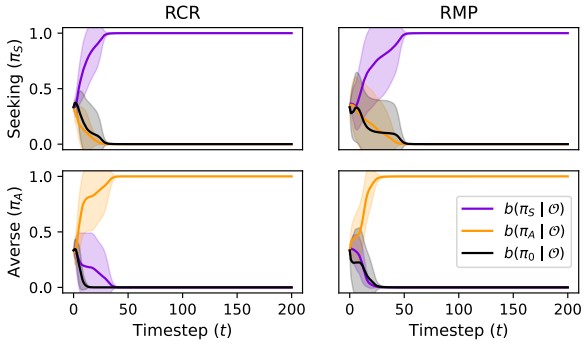

*Figure 4.* Bayesian inference $b(\pi \mid \mathcal{O})$ converging to correct belief about a risk-seeking $\pi_S$ (top row) or risk-averse $\pi_A$ (bottom row) partner in RCR and RMP layouts.

**Team Performance.** On average across both risk-sensitive partners, an RS-ToM improved upon the reward $\Sigma r$ of RAT by a factor $+77\%$ in RCR and $+189\%$ in RMP. In RCR, gains were more apparent with an Averse partner ($+109\%$) than a Seeking one ($+53\%$) while the inverse is true in RMP with a percent gain of $+127\%$ and $+273\%$ for Averse and Seeking partners, respectively. The discrepancy in RCR is primarily caused by the risk/rely framing where failure to pass objects through the center counter space (A and B in Fig 3) comes at a high cost which affords greater opportunity for performance gains with an RS-ToM. Discrepancy in RMP can be attributed to added congestion of risk-seeking paths due to little utilization of the open $2 \times 3$ space that allows partners to pass each other (see G in Fig 3). Post-hoc testing revealed a non-significant ($p = 0.06$) difference in rewards between an Averse and Seeking partner with an RS-ToM in RCR but a significant difference in RMP.

**Coordination Fluency.** On average across all experiments, an RS-ToM improved upon RAT by a factor of $104\%$. Gains for both partners were larger in RMP ($+150\%$) than in RCR

($+59\%$) which is best attributed as a product of the layout design. Notably, there was larger gains in RCR with an Averse partner ($+81\%$) than with a Seeking one ($+44\%$) due to the Averse strategy attempting to avoid puddles through coordinated handoffs to the partner (A and B in Fig 3) that are never completed in RAT. Thus, the Averse partner is left with high inactivity when waiting for this to occur.

**Risk Taking Alignment.** The number of risks taken by the team $N_{risk}$ shows that an RS-ToM allows the risk-sensitive partner to follow their preferred risk-taking tendencies. This can be seen by the near identical $N_{risk}$ between RS-ToM and Oracle (i.e. the ideal risk-taking tendency to the partner). In contrast to Oracle, RAT decreased $N_{risk}$ when interacting with a Seeking partner and increased $N_{risk}$ with an Averse partner. The only exception is for RCR with a Seeking partner which is attributed to discoordination in a confined space requiring backtracking into puddles in a way not present in RMP.

**Predictability.** The predictability of the risk-sensitive partner $\mathcal{P}_H$ over doubled ($+121\%$) with an RS-ToM on average across all experiments. In RCR, RAT showed more difficulty predicting an Averse partner's action while the Seeking partner is more difficult to predict in RMP. Here, $\mathcal{P}_H$ saw $+81\%$ and $+188\%$ increase for Seeking and Averse partners, respectively, in RCR and a $+147\%$ and $+97\%$ increase in RMP. There were non-significant post-hoc test comparisons between Seeking and Averse partners when using RS-ToM in both the RCR ($p = 0.42$) and RMP ($p = 0.82$) layouts. This is attributed to equal decision temperatures $\lambda$ for Seeking and Averse policies once the correct policy was inferred.

## 4. Discussion

The experiments with an RS-ToM show the ability to learn diverse, risk-sensitive behaviors in high-dimensional tasks and the ability to correctly adapt to an agent of unknown risk-sensitivity online. This enables us to build upon prior works that support risk-sensitive human interaction by affording interaction more realistic scenarios (i.e., complex tasks and personalization without a data prior). As CPT parameters describe different biases, this also provides us with an interpretable method for generating arbitrary risk-sensitive behavior in multi-agent, risky decision settings.

We then validate the RS-ToM by comparing it to the conventional assumption of noisy-rationality. We evaluated these algorithms in two different cases (i.e., interacting with risk-seeking or risk-averse partner) where the partner's risk-sensitivity was unknown to the agent. Results ubiquitously show that interaction can be improved with RS-ToM and that the standard assumption of rationality in RAT is insufficient for effective coordination to occur. The near identical performance of an RS-ToM and Oracle indicates that the inference step was able to quickly converge to the correct belief about partner risk-sensitivity. Thus, the RS-ToM was able to recover the optimal policy effectively allowing comparable performance to the ideal strategy.

While these results are performed with an artificial partner, insights on the consequences of an RS-ToM when interacting with real humans may be extrapolated. Similar transfer between simulation and human studies is seen in (Smith & Zhang, 2025), albeit with a less dramatic effect due to the human's ability to adapt to misaligned risk-sensitive models. The results showed that RAT tended to cause risk taking strategies that were contrary to the partner's risk preferences (i.e., Averse saw an increase in $N_{risk}$ while Seeking saw a decrease) whereas RS-ToM was able to quickly infer and align to its partner's preferences to avoid this. Consequently, an RS-ToM would likely avoid deploying strategies that could be perceived as inappropriate (i.e., "too risky" or "too safe") in a way that could damage interpersonal features like trust (Lee & Moray, 1992). The improvements to coordination fluency with an RS-ToM is also likely to promote appreciation and confidence by the human (Hoffman, 2019). Thus, we suspect that an agent with an RS-ToM would be perceived as a better teammate.

**Limitations and Future Work.** The RCR and RMP layouts were selected to maximize diversity across framings. However, the observed context-dependent effects may suggest that there are unobserved risk-sensitive behaviors that may emerge with novel layout designs. Moreover, although RCR and RMP were sufficient for demonstrating the RS-ToM algorithm in simulation, future studies with real human partners may want to deploy a wider variety of layouts to ensure that their unique risk-sensitivity has the opportunity to present itself in a significant way.

While we believe that modeling human risk-sensitivity is an essential factor in risky settings, such behaviors could feasibly be described by alternative human models like limited planning horizon due to the discrete set of strategies in Risky Overcooked (i.e., traverse puddle, take a detour, or handoff object). This overlap will likely fade as biases become more differentiable either through performance of multiple Risky Overcooked layouts or with increased decision complexity. However, a multi-modal approach (Kryven et al., 2024) to modeling human bias may be an valuable future direction.

However, the discrete set of Risky Overcooked strategies is helpful in clearly showing RS-ToM's ability to adapt to risk-sensitive partners, as three candidate policies are sufficient for describing these high-level behaviors. Nevertheless, increasing the resolution of the candidate policies may be helpful for scaling to tasks with more a diverse set of risky-decisions (e.g., continuous control problems affording continuous variation in risk-sensitive strategies). Although an RS-ToM can be arbitrarily scaled with additional candidate policies as the task demands, meta-model methods that can continuously interpolate between candidates (Hong et al., 2023) or transfer learning (Weiss et al., 2016) may be a promising direction for increasing RS-ToM resolution.

## 5. Conclusion

Standard approaches to HAT model the human as being noisy-rational which can lead to poor coordination when they are subject to risk-sensitive biases. To address this, our work presents a novel RS-ToM framework that generates personalized adaption to a partner of unknown risk-sensitivity without the need for a data prior. In a simulated study, we show that an RS-ToM is able to better align itself with its partner's risk preferences and coordinate its behavior in a way that significantly outperforms noisy-rationality. Thus, these results imply that an agent endowed with an RS-ToM would be able to facilitate more effective teaming when paired with real humans.

## Software and Data

Training of policies used in the RS-ToM was conducted in Python 3.11 using PyTorch 2.3.1's GPU acceleration. Implementation of the Risky Overcooked environment, the RS-ToM algorithm, and pre-trained models can be found at `https://github.com/ASU-RISE-Lab/risky_overcooked/`.

## Acknowledgements

This material is based upon work supported by the Air Force Office of Scientific Research under Award No. FA9550-23-1-0283.

## Impact Statement

This paper presents work whose goal is to advance the field of human-aware reinforcement learning. There are many potential societal consequences of our work, most important of which is machines that understand biased humans so that they can effectively predict, coordinate, and maintain trustworthy relationships in the real world.

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

# A. Implementation

## A.1. State and Action Space

The learning problem in Overcooked is commonly simplified in two ways (Carroll et al., 2019). First is by using mid-level action planners that allow agents to make decisions over high-level actions such as get an onion and serve the dish, and rely on algorithms such as $A^*$ search to solve the low-level routing problem. The problem is further simplified by creating a custom state featurization that specifies a state with distance $(dx, dy)$ to the *closest* onion, dish, soup, and their dispensers. Therefore, we lose information on all other objects in the environment. Both of these approaches are incompatible with Risky Overcooked as the low-level routing problem that $A^*$ solves is where the risky decision is present (e.g., traverse puddle or take a detour). Also, we need full observability of all objects in the environment as an risk-sensitive agent may want to navigate to an object that is not the closest if a puddle is in the way.

We define the action space for each agent as $A_k := \{up, down, left, right, interact, wait\}$ where the size of the joint-action space is $|A| = 36$. Also, we formulate a lossless state encoding vector to enable full observability of the environment. This first consists of features describing each agent: location $(x, y)$, a one-hot encoding of orientation $\{up, down, left, right\}$, and a one-hot encoding of held object $\{onion, dish, soup\}$. As a result, the total length of the feature vector describing both agents is 18.

Next, we have to encode objects that may be placed on any reachable counter space. Thus, every counter is given a one-hot encoding of $\{onion, dish, soup\}$ resulting in a total feature vector length of $3c$ where $c$ is the number of reachable counters.

Lastly, we encode the status of each pot with a single value where we assign sequential pot status as $\{0 : empty\ pot,\ 1 : contains\ one\ onion, 2 : contains\ two\ onions, 3 : contains\ three\ onions, 4 : soup\ is\ ready\}$. Thus, the total length of the feature vector is $z$ where $z$ is the number of pots.

This lossless state encoding varies with the layout where the total length can be expressed as $|s| = 18 + 3c + z$. This implies that there may be scalability issues in extremely large environments, especially as more open counter spaces are added. For the RCR and RMP layouts, we have $|s| = 41$ and $|s| = 68$, respectively.

## A.2. Handling Sparse Rewards

Risky Overcooked is an even more sparsely rewarded environment than the original Overcooked task due to the possibility of losing held objects. To address this during learning, we first apply a naive curriculum that consisted of learning the following subtasks in order: 1) deliver the soup, 2) pick up the cooked soup from the pot, 3) grab the dish before picking up the soup, 4) pot the third onion, 5) pot the second onion, 6) pot the first onion, and 7) full task. This curriculum is equivalent to having the agents learn the task backwards such that the steps for obtaining the primary payoff $r = 20$ for delivering the soup only requires one subtask to be learned where later subtasks are already solved. During sampling, a decaying probability of sampling previous curricula was added so subtask strategies were not forgotten.

Also, a series of small, intermediate rewards are provided to agents upon completion of subtasks. This helps reduce the sparsity of rewards in the environment and speeds up convergence of the MARSRL algorithm. A linearly decaying weight is added to these shaping rewards that converges to 0 as training progresses. When an agent places an onion in a pot or picks up a dish when a pot contains a finished soup, the agent will receive a $\tilde{r} = 3$ reward. When an agent successfully picks up a soup by bringing a plate to a pot with a finished soup, they receive a $\tilde{r} = 5$ reward. As we attempt to explore behaviors involving reliance and coordinated handoffs (i.e., risk/rely framing), we do not want to only reward the agent that completes the subtasks and encourage greedy and independent behaviors. Conversely, if agents shared all shaped rewards, an agent that did not participate in a subtask would receive frivolous reward signals that do not support effective learning. Thus, an agent receives a shaped reward only if they participated in that subtask. For example, both agents will receive the shaped reward if $k$ picks up an onion and places it on an open counter so -$k$ can pick it up and place it in the pot. However, $k$ is required to have interacted with the object relevant to the completed subtask at some point in order to be rewarded.

## A.3. Hyperparameter Selection

An exhaustive grid search over the values in Table 2 was used to optimize model hyperparamters. The search was ran with a rational agent in the RCR layout and final values were selected based on the model with the maximum mean reward over the past 10 iterations to select for both optimality and stability of the policy. Selected hyperparamters used during training of models are indicated by the bold values in Table 2.

*Table 2.* Hyperparamters Tuning and Selection

| Hyperparameter | Value(s) |
|---|---|
| Discount factor ($\gamma$) | $\{\mathbf{0.95}, 0.97, 0.99\}$ |
| Learning rate ($\alpha$) | $\{5e\text{-}3, \mathbf{1e\text{-}3}, 5e\text{-}4\}$ |
| Target soft update rate ($\sigma$) | $\{\mathbf{5e\text{-}3}, 1e\text{-}3, 5e\text{-}4\}$ |
| Minibatch size ($N$) | $\{128, \mathbf{256}\}$ |
| Num. hidden layers | $\{3, \mathbf{5}, 7\}$ |
| Hidden layer size | $\{\mathbf{128}, 256\}$ |
| Memory size | $\{2e4, 5e4, \mathbf{1e5}\}$ |

## A.4. Quantal Response Equilibrium

A QRE (McKelvey & Palfrey, 1995) is a game theoretic solution to multi-agent choice problems. Here, players choose strategies using the expected utility of that strategy and assume that other players do the same. In other words, both players' payoffs are computed conditionally on their beliefs about the probability distribution of the other player's actions. This creates a recursive set of beliefs where the QRE is obtained when utility functions converge as the level of recursion goes to $\infty$. Similarly, the QRE converges to a Nash equilibrium solution as the decision temperature $\lambda \to \infty$ (Goeree et al., 2020). However, humans do not have the resources to compute this infinite sequence due to bounded rationality. Therefore, we apply a $\varphi$-level reasoning paradigm where we terminate the recursive sequence after $\varphi$ iterations. While traditionally refereed to as "level-k" reasoning, we use $\varphi$ instead to avoid confusion with agent indexes $k$ or $\text{-}k$. When $\varphi = 0$, we assume that a player has a uniform probability distribution $\mathbf{U}$ over their controllable actions. Consequently, if $\varphi = 3$, we get the following sequence of calculations:

$$p_k^{\varphi=3}(A_k) \propto \mathbb{E}[Q(s, a_k) | p_{\text{-}k}^{\varphi=2}(A_{\text{-}k})],$$
$$p_{\text{-}k}^{\varphi=2}(A_{\text{-}k}) \propto \mathbb{E}[Q(s, a_{\text{-}k}) | p_k^{\varphi=1}(A_k)],$$
$$p_k^{\varphi=1}(A_k) \propto \mathbb{E}[Q(s, a_k) | p_{\text{-}k}^{\varphi=0}(A_{\text{-}k})],$$
$$p_{\text{-}k}^{\varphi=0}(A_{\text{-}k}) = \mathbf{U},$$

Thus, an equilibrium strategy $(\pi_k(s), \pi_{\text{-}k}(s))$ to a single stage game $Q(s)$ can be computed where we assume that $\varphi_k = \varphi_{\text{-}k}$.

## A.5. Convergence Proof

In order to establish convergence of Algorithm 1, we will first prove that rational (non-CPT) updates performed on the Q-values in (6) under a QRE solution constitute a contraction mapping and that $Q^*_{cpt}$ is a fixed-point. This proof follows closely with (Hu & Wellman, 2003) with minor modifications to the equilibrium solution and formulation as a two-player game. Once this proof is concluded, an extension to the convergence of the risk-sensitive version in (8) will be demonstrated using the results in (Lin et al., 2018).

We first rely on two assumptions about infinite sampling and decaying of learning rate. Also, we assume that each stage of the game is structured such that an optimal solution can be consistently found.

**Assumption A.1.** Each joint state-action pair is observed infinitely often

**Assumption A.2.** The learning rate $\alpha$ satisfies $\alpha \in [0, 1)$, $\sum_{t=0}^{\infty} \alpha_t = \infty$, and $\sum_{t=0}^{\infty} \alpha_t^2 < \infty$.

**Assumption A.3.** Every stage $t$ game $Q_t(s) \ \forall \ t, s$ has a globally optimal point[7].

Assumption A.3 may seem strong at first glance when using QRE as it can converge to an arbitrary Nash equilibrium solution, and not necessarily the global optima, as the decision temperature $\lambda \to \infty$ (Goeree et al., 2020). However, we support Assumption A.3 with the fact that Risky Overcooked is a common-payoff game and that the QRE will most frequently converge to the globally optimal point in this case. To simplify notation, we denote the $\pi_k(s')\pi_{\text{-}k}(s')Q(s')$ as shorthand for $\sum_{a'_k, a'_{\text{-}k}} \pi_k(a'_k \mid s')\pi_{\text{-}k}(a'_{\text{-}k} \mid s')Q(s', a'_k, a'_{\text{-}k})$ that describes the value for agent $k$ as a result of the QRE solution $(\pi_k(s'), \pi_{\text{-}k}(s'))$ to the stage game $Q(s')$.

---

[7]In (Hu & Wellman, 2003), the authors assume that every stage game may also be a saddle point which relaxes this assumption. However, the common-payoff structure of Risky Overcooked prevents saddle points from existing.

**Definition A.4.** A joint strategy $(\pi_k^*(s), \pi_{-k}^*(s))$ of stage game $Q_t(s)$ is a global optimal point if every agent receives their highest payoff at this point. That is,

$$\pi_k^*(s)\pi_{-k}^*(s)Q_t(s) \geq \pi_k(s)\pi_{-k}(s)Q_t(s) \ \forall \ \pi_k, \pi_{-k} \in \Pi$$

**Definition A.5.** A common-payoff game is a game in which for all joint strategies $(\pi_k(s), \pi_{-k}(s))$, it is the case that agents receive the same payoff such that $\pi_k(s)\pi_{-k}(s)Q_k(s) = \pi_k(s)\pi_{-k}(s)Q_{-k}(s)$

By Definition A.5, we have $\pi_k^*(s)\pi_{-k}^*(s)Q_k(s) = \pi_k^*(s)\pi_{-k}^*(s)Q_{-k}(s)$ which satisfies Definition A.4 and guarantees the existence of a globally optimal point in common-payoff games.

**Proposition A.6.** *Given the existence of a globally optimal point and a QRE that terminates at a level-0 agent with a uniform strategy $\pi_U$, the globally optimal point is the most likely equalibrium solution as $\lambda \to \infty$.*

*Proof.* Consider a uniform stage game $Q(s, a_k, a_{-k}) = q \ \forall \ a_k, a_{-k}$ except at the global optima $a_k^*, a_{-k}^*$ such that $Q(s, a_k^*, a_{-k}^*) > q$. Then, given that $-k$ is a level-0 agent in a QRE and assumed to have a uniform strategy $\pi_U$ by $k$, we have $\sum_{a_{-k}} \pi_U(a_{-k} \mid s)Q(s, a_k^*, a_{-k}) \geq \sum_{a_{-k}} \pi_U(a_{-k} \mid s)Q(s, a_k, a_{-k}) \ \forall \ a_k$ making the QRE guaranteed to select the globally optimal action $a_k^*$ as $\lambda \to \infty$ since $\pi_k(a_k \mid s) \propto exp(\lambda \sum_{a_{-k}} \pi_U(a_{-k} \mid s)Q(s, a_k, a_{-k}))$. Subsequent level-k iterations will continue to select the pure strategy $(a_k^*, a_{-k}^*)$ as this is a strictly dominant strategy given this initial condition. If we instead let all all other values of $Q(s, a_k, a_{-k})$ be randomly sampled from a distribution with a mean of $q$, we expect the same to hold for most cases, especially when $Q(s, a_k^*, a_{-k}^*) \gg q$. A similar argument holds for mixed equilibrium strategies where the global optima $(\pi_k^*, \pi_{-k}^*)$ is the most probable solution. In the case that $(\pi_k^*, \pi_{-k}^*)$ is a strictly dominated strategy, the globally optimal solution is guaranteed to be selected.

Thus, we support the Assumption A.3 with the fact that QRE is more likely to converge to a global optima rather than other Nash equilibrium solutions with a sufficiently high $\lambda$ in an ambiguous case.

The majority of the remaining proof will be working towards showing that (6) meets the conditions of the following theorem to provide a means for showing convergence.

**Theorem A.7.** *Let the mapping $\mathcal{T}_t : \mathbb{Q} \to \mathbb{Q}$ satisfy the following condition: there exists a $0 \leq \gamma < 1$, such that $||\mathcal{T}_t Q \leq \mathcal{T}_t \hat{Q}|| < \gamma ||Q - \hat{Q}|| \ \forall \ Q, \hat{Q} \in \mathbb{Q}$ and $\mathcal{T}_t$ admits a unique fixed point $Q^* \in \mathbb{Q}$ such that $Q^* = \mathbb{E}[\mathcal{T}_t Q^*]$. Then, the Q-iteration operator $(\mathcal{T}_t Q)(s, a_k, a_{-k}) = r(s, a_k, a_{-k}) + \gamma \pi_k(s')\pi_{-k}(s')Q(s')$ performed during the Q-value update converges to $Q^*$ with probability 1.*

where $\mathbb{Q}$ is the space of all possible Q-functions. Here, we can equivalently define $(\mathcal{T}_t Q)(s, a_k, a_{-k}) = r(s, a_k, a_{-k}) + \gamma V(s' \mid \pi_k, \pi_{-k})$. The simplest condition of Theorem A.7 to show is that $Q^*$ is a fixed point.

**Lemma A.8.** *For a two-player stochastic game, $\mathbb{E}[\mathcal{T}_t Q^*] = Q^*$*

*Proof.* Following from (5), with an equilibrium $(\pi_k(s), \pi_{-k}(s))$ on a shared optimal Q-function $Q^*$, we have

$$Q^*(s, a_k, a_{-k}) = r(s, a_k, a_{-k}) + \gamma \sum_{s' \in S} \mathbb{P}(s' \mid s, a_k, a_{-k})V^*(s' \mid \pi_k, \pi_{-k})$$

$$= r(s, a_k, a_{-k}) + \gamma \sum_{s' \in S} \mathbb{P}(s' \mid s, a_k, a_{-k})\pi_k(s')\pi_{-k}(s')Q^*(s')$$

$$= \sum_{s' \in S} \mathbb{P}(s' \mid s, a_k, a_{-k})\Big(r(s, a_k, a_{-k}) + \gamma \pi_k(s')\pi_{-k}(s')Q^*(s')\Big)$$

$$= \mathbb{E}[\mathcal{T}_t Q^*(s, a_k, a_{-k})]$$

for all $s, a_k, a_{-k}$. Thus, $\mathbb{E}[\mathcal{T}_t Q^*] = Q^*$.

We must now prove that $\mathcal{T}_t$ is a contraction which requires that every stage game encountered during learning has a unique solution and that the learner consistently chooses this solution when updating its Q-values (i.e., Assumption A.3).

We then define the distance between two Q-functions.

**Definition A.9.** For $Q, \hat{Q} \in \mathbb{Q}$, define

$$||Q - \hat{Q}|| \equiv \max_s ||Q(s) - \hat{Q}(s)||$$

$$\equiv \max_s \max_{a_k, a_{-k}} |Q(s, a_k, a_{-k}) - \hat{Q}(s, a_k, a_{-k})|$$

Given Assumption A.3, we establish that $\mathcal{T}_t$ is a contraction mapping.

**Lemma A.10.** $||\mathcal{T}_t Q - \mathcal{T}_t \hat{Q}|| \leq \gamma ||Q - \hat{Q}|| \ \forall \ Q, \hat{Q} \in \mathbb{Q}$

*Proof.* For any Q-functions $Q, \hat{Q} \in \mathbb{Q}$

$$||\mathcal{T}_t Q - \mathcal{T}_t \hat{Q}|| = \max_s |\gamma \pi_k(s) \pi_{-k}(s) Q(s) - \gamma \hat{\pi}_k(s) \hat{\pi}_{-k}(s) \hat{Q}(s)|$$

$$= \max_s \gamma |\pi_k(s) \pi_{-k}(s) Q(s) - \hat{\pi}_k(s) \hat{\pi}_{-k}(s) \hat{Q}(s)|$$

We then seek to prove that

$$|\pi_k(s) \pi_{-k}(s) Q(s) - \hat{\pi}_k(s) \hat{\pi}_{-k}(s) \hat{Q}(s)| \leq ||Q(s) - \hat{Q}(s)||$$

Consider that both $(\pi_k(s), \pi_{-k}(s))$ and $(\hat{\pi}_k(s), \hat{\pi}_{-k}(s))$ satisfy Assumption A.3 by being the global optimal solutions over $Q$ and $\hat{Q}$, respectively. If $\pi_k(s)\pi_{-k}(s)Q(s) \geq \hat{\pi}_k(s)\hat{\pi}_{-k}(s)\hat{Q}(s)$, we have

$$\pi_k(s)\pi_{-k}(s)Q(s) - \hat{\pi}_k(s)\hat{\pi}_{-k}(s)\hat{Q}(s)$$

$$\leq \pi_k(s)\pi_{-k}(s)Q(s) - \pi_k(s)\pi_{-k}(s)\hat{Q}(s)$$

$$= \sum_{a_k, a_{-k}} \pi_k(s, a_k)\pi_{-k}(s, a_{-k})(Q(s, a_k, a_{-k}) - \hat{Q}(s, a_k, a_{-k}))$$

$$\leq \sum_{a_k, a_{-k}} \pi_k(s, a_k)\pi_{-k}(s, a_{-k}) \max_{a_k, a_{-k}} |Q(s, a_k, a_{-k}) - \hat{Q}(s, a_k, a_{-k})|$$

$$= \sum_{a_k, a_{-k}} \pi_k(s, a_k)\pi_{-k}(s, a_{-k})||Q(s) - \hat{Q}(s)||$$

$$= ||Q(s) - \hat{Q}(s)||$$

If $\pi_k(s)\pi_{-k}(s)Q(s) \leq \hat{\pi}_k(s)\hat{\pi}_{-k}(s)\hat{Q}(s)$, then $\pi_k(s)\pi_{-k}(s)Q(s) - \hat{\pi}_k(s)\hat{\pi}_{-k}(s)\hat{Q}(s) \leq \hat{\pi}_k(s)\hat{\pi}_{-k}(s)Q(s) - \hat{\pi}_k(s)\hat{\pi}_{-k}(s)\hat{Q}(s)$ which affords a similar proof to the above. Thus, $\mathcal{T}_t$ is a contraction mapping under the given assumptions.

**Theorem A.11.** *Under Assumptions A.1-A.3, the sequence $Q_t$ updated by:*

$$Q_{t+1}(s, a_k, a_{-k}) = (1 - \alpha)Q_t(s, a_k, a_{-k}) + \alpha \Big( r_t + \gamma \pi_k(s')\pi_{-k}(s')Q_t(s') \Big)$$

*where $(\pi_k(s'), \pi_{-k}(s'))$ is the level-k quantal response equilibrium solution for a stage game $Q_t(s)$, converges to the Q-value $Q^*$.*

*Proof.* Proof of convergence to $Q^*$ follows directly from Theorem A.7 which requires that $\mathcal{T}_t$ is a contraction operator, established in Lemma A.10, and that the fixed-point condition $\mathbb{E}[\mathcal{T}_t Q_t] = Q^*$ is satisfied according to Lemma A.8. Therefore, the process $(1 - \alpha_t)Q_{t+1} + \alpha_t[\mathcal{T}_t Q_t]$ converges to $Q^*$.

We will now extend the previous proof of convergence of the rational (non-CPT) algorithm to its risk-sensitive variant.

**Theorem A.12.** *Let the utility weighting functions $u^+$ and $u^-$ be according to Definition 2.1 and assume they are invertible, differentiable, and have monotonically non-increasing derivatives. Also, let probability weighting functions $w^+$ and $w^-$ be according to Definition 2.2 and monotone. Then, the following is satisfied $||\mathcal{T}_t Q_{cpt} - \mathcal{T}_t \hat{Q}_{cpt}|| < \gamma ||Q_{cpt} - \hat{Q}_{cpt}|| \ \forall \ Q_{cpt}, \hat{Q}_{cpt} \in \mathbb{Q}_{cpt}$.*

*Proof.* This proof follows similarly to (Ramasubramanian et al., 2021) where we leverage the fact that utility weighting functions are monotonically non-decreasing (Definition 2.1). Thus, given a policy is improved such that $\pi'_k(s)\pi'_{-k}(s')Q_{cpt}(s) \geq \pi_k(s)\pi_{-k}(s)Q_{cpt}(s)$, then we have

$$u^+(r(s, a_k, a_{-k}) + \gamma \pi'_k(s')\pi'_{-k}(s')Q_{cpt}(s') \geq u^+(r(s, a_k, a_{-k}) + \gamma \pi_k(s')\pi_{-k}(s')Q_{cpt}(s')) \tag{10}$$

where we denote the above inequality as $u^+_{\pi'} \geq u^+_\pi$. Since the probability weighting functions in Definition 2.2 are also monotonically non-decreasing, we can then write the continuous form of the calculation of the CPT-value in (3) as

$$\int_0^\infty w^+(\mathbb{P}(u^+_{\pi'} > z))dz \geq \int_0^\infty w^+(\mathbb{P}(u^-_\pi > z))dz \tag{11}$$

where a similar argument holds for $u^-$ and $w^-$. Thus, the operator $\mathcal{T}_t Q_{cpt}$ is monotone. We can then show that

$$
\begin{aligned}
(\mathcal{T}_t & Q_{cpt})(s, a_k, a_{-k}) \\
&= \rho_{cpt}\Big(r(s, a_k, a_{-k}) + \gamma \pi_k(s')\pi_{-k}(s')Q_{cpt}(s)\Big) \\
&= \rho_{cpt}\Big(r(s, a_k, a_{-k}) + \gamma \sum_{a_k, a_{-k}} \pi_k(s', a_k)\pi_{-k}(s', a_{-k})(\hat{Q}_{cpt}(s') + Q_{cpt}(s') - \hat{Q}_{cpt}(s'))\Big) \\
&\leq \rho_{cpt}\Big(\gamma \epsilon + r(s, a_k, a_{-k}) + \gamma \sum_{a_k, a_{-k}} \pi_k(s', a_k)\pi_{-k}(s', a_{-k})\hat{Q}_{cpt}(s')\Big)
\end{aligned}
$$

where $\epsilon = ||Q(s') - \hat{Q}(s')||$. We then rely on Theorem 6 in (Lin et al., 2018) and its proof that provides an analysis of integrals composing $\rho_{cpt}(\cdot)$ to obtain $(\mathcal{T}_t\hat{Q})(s, a_k, a_{-k}) \leq (\mathcal{T}_t\hat{Q})(s, a_k, a_{-k}) + \gamma\epsilon$ which results in $||\mathcal{T}_t Q - \mathcal{T}_t\hat{Q}|| \leq \gamma||Q - \hat{Q}|| \; \forall \, Q, \hat{Q} \in \mathbb{Q}$ as a consequence. Thus, we have shown that both the QRE and CPT-value modifications constitute a contraction mapping. This concludes the proof.

### A.5.1. Upper-Bound on $\ell$

It is helpful here to mention a potential challenge when choosing CPT parameters in a way that ensures convergence. Specifically, the loss aversion parameter $\ell$ introduces a gain on the TD-target which may introduce a means for the Q-function to diverge under certain conditions. This section will provide a heuristic upper-bound on $\ell$ such that we can maintain the critical assumption that $\gamma < 1$ in Theorem A.7.

Consider the worst-case scenario where every TD-Target prospect $\tau_i = r(s, a) + \gamma \sum_{s'} V(s')$ with probability $p_i = \mathbb{P}(s' \mid s, a)$ is perceived as a loss (i.e. $\tau_i \leq b \; \forall \, i$). Thus, the loss's utility function $u^-(\tau_i) = \ell|\tau_i - b|^{\eta^-}$, where we assume $b = 0$ for simplicity, can be expressed as:

$$
\begin{aligned}
u^-(\tau_i) &= \ell|\tau_i|^{\eta^-} \\
&= \ell|r(s, a) + \gamma V(s' \mid \pi_k, \pi_{-k})|^{\eta^-} \\
&= |\ell^{\frac{1}{\eta^-}}r(s, a) + \ell^{\frac{1}{\eta^-}}\gamma V(s' \mid \pi_k, \pi_{-k})|^{\eta^-}
\end{aligned}
$$

Letting $\hat{\gamma}_0 = \ell^{\frac{1}{\eta^-}}\gamma$, we can satisfy the same condition for convergence in the worst-case scenario given that $\hat{\gamma}_0 < 1$. For example, we would have $\hat{\gamma}_0 = 2.23$ with $\ell = 2.25$, $\eta^- = 1$, and $\gamma = 0.99$. This would provide no means for future values to decay where the Q-function would quickly grow towards $-\infty$ or $\infty$. However, given that $\tau_i$ can be perceived as a gain or a loss, this constraint on parameters $\ell$ and $\eta^-$ may be relaxed but there is no theoretical grantees since the expected $\hat{\gamma}$ would need to be calculated from the cumulative CPT expectation in (3). A naive estimate can be made with

$$
\hat{\gamma}_1 \approx \frac{l}{K}\hat{\gamma}_0 + \frac{K - l}{K}\gamma
$$

where $l$ is the number of TD-targets perceived as a loss and $K$ is the number perceived as gains. There is no feasible method for determining $\frac{l}{K}$ theoretically before algorithm runtime. However, this motivates the use of the *pragmatic CPT-value* that only applies CPT expectation during choices that involve a risky outcomes where $|\tau| > 1$. This further relaxes the constraints on $\ell$ and $\eta^-$ since the

$$
\hat{\gamma}_2 \approx \frac{n_\rho}{T}\hat{\gamma}_1 + \frac{T - n_\rho}{T}\gamma
$$

where $\frac{n_\rho}{T}$ is the proportion of risky decisions under some time horizon $T$. Then, if $\hat{\gamma}_2 < 1$ as $T \to \infty$, the condition for the Bellman equation should be met. Given the sparsity of risky-decisions in Risky Overcooked and reasonable choices of $\ell$ and $\eta^-$, this condition is almost surely satisfied.

## B. Experiment Details

### B.1. Risk-Sensitive Parameter Selection

The goal of parameter selection is to create sufficient divergence in risk-sensitive strategies (e.g., traverse risky states versus passing objects through the center counter in RCR) while still being cognitively motivated. Therefore, we take the median

CPT parameter estimates in (Tversky & Kahneman, 1992) and fine tune them to obtain clear definitions for risk-averse and risk-seeking candidate policies. A summary of the selected parameters can be found in Table 3 while the prospect curves illustrating these parameters can be found in Fig. 5. The remainder of this section will describe the rationale behind these selections.

*Table 3.* CPT Parameter Selection

| Policy | Utility Function | | | Weighting Function | |
|---|---|---|---|---|---|
| | $\ell$ | $\eta^+$ | $\eta^-$ | $\delta^+$ | $\delta^-$ |
| Risk-Seeking | 0.44 | 1 | 0.88 | 0.61 | 0.69 |
| Risk-Neutral | 1 | 1 | 1 | 1 | 1 |
| Risk-Averse | 2.25 | 0.88 | 1 | 0.61 | 0.69 |

For the loss-aversion parameter, the median estimate is $\ell = 2.25$ where $\ell > 1$ creates risk-averse tendencies while $\ell < 1$ creates risk-seeking tendencies. Therefore, we select $\ell = 2.25$ for the risk-averse candidate policy and its inverse $\ell = 0.44$ for the risk-seeking candidate policy. For the exponential utility discounting parameters, a $\eta^+ < 1$ will generate risk-averse tendencies while $\eta^- < 1$ generates risk-seeking tendencies. Given that the median estimates were found to be $\eta^+ = \eta^- = 0.88$, we can select $\eta^+ = 0.88$ and $\eta^- = 1$ such that there is exponential discounting of gains and none on losses to emulate an agent perceiving gains as less important than losses. This results in a consistently risk-averse utility transformation in Fig. 5 across all possibly utility values. Similarly, we select $\eta^+ = 1$ and $\eta^- = 0.88$ to emulate perceptions that weights losses as less important than gains. This generate the risk-seeking utility transformation in Fig. 5. By selecting the nominal value of 1 for the $\eta$ that would generate the opposite of the intended risk-sensitive behaviors, in conjunction with proper selection of $\ell$, we can ensure that there is no conflicting utility weightings that would otherwise interfere with a clear definition of risk-averse or risk-seeking behaviors.

The interpretation of the probability weighting parameters $\delta^+$ and $\delta^-$ is context dependent and leads to a infeasible analysis across tasks (i.e. $p_\rho$ changes between layouts). Therefore, the median estimates of $\delta^+ = 0.61$ and $\delta^- = 0.69$ were used identically for both risk-averse and risk-seeking agents in the decision weighting transformation in Fig. 5. To obtain a noisy-rational candidate policy, we simply set all CPT parameters to their nominal value of 1 that creates an identity transformation for all functions.

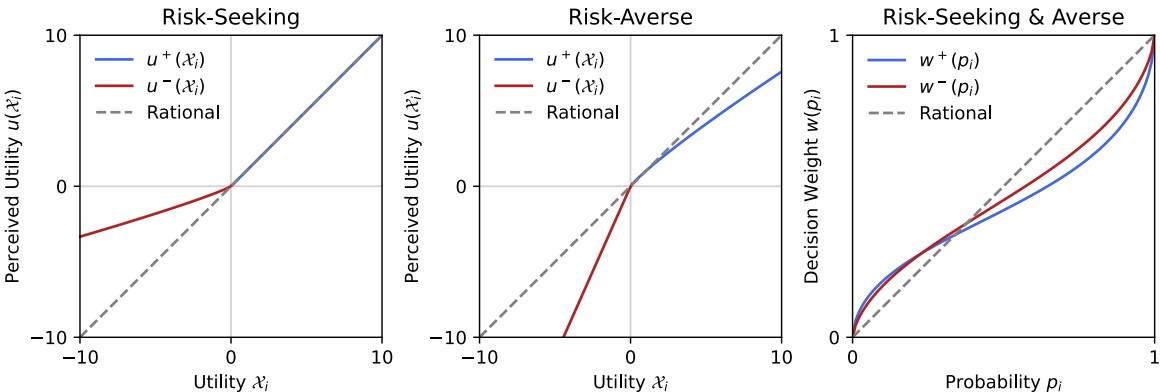

*Figure 5.* Prospect curves for the selected CPT parameters used to generate the risk-seeking $\pi_S$ and risk-averse $\pi_A$ candidate policies. The prospect curves in all plots for the risk-neutral $\pi_0$ candidate policy is show by the black dashed line in all figures and is equivalent to the identity transformation.

### B.2. Tuning Risk Levels

The probability of slipping $p_\rho$ (i.e. risk level) was tuned to maximize *mutual* dissimalarity between the risk-seeking $\pi_S$, risk-averse $\pi_A$, and risk-neutral (i.e. rational) $\pi_0$ policies. We specify "mutual" here to express the desire to weight cases where a $p_\rho$ generates one extremely dissimilar policy and two similar ones less than a $p_\rho$ that generates three equally dissimilar policies with a possibly lower cumulative dissimilarity measure. Doing so allows us to optimize each layout to

amplify the divergence of the tested candidate policies, perform better inference, and generate more insightful experiments.

We implement the Jensen–Shannon divergence (JSD) (Nielsen, 2019) as our similarity measure which is a symmetrized version of the Kullback–Leibler divergence measure where the JSD between policies $\pi_1$ and $\pi_2$ can be expressed as:

$$D_{JS}(\pi_1 \| \pi_2) = \frac{1}{2} D_{KL}(\pi_1 \| M) + \frac{1}{2} D_{KL}(\pi_1 \| M) \tag{12}$$

where $M = \frac{1}{2}(\pi_1 + \pi_2)$ is the mixture distribution and $D_{KL}$ is the Kullback–Leibler divergence between $\pi$ and $M$. For each of the three combinations of policies $\{(\pi_S, \pi_A), (\pi_S, \pi_0), (\pi_A, \pi_0)\}$, $D_{JS}(\pi_1 \| \pi_2) \in \mathbf{D}$ is computed. To maximize *mutual* dissimilarity, additional JSD beyond the minimum is discounted:

$$\sigma = \sum_{d \in \mathbf{D}} log(d - \min(\mathbf{D}) + 1) + \min(\mathbf{D}) \tag{13}$$

where $\sigma \in [0, 1]$ is a measure of mutual dissimilarity and $\sigma = 0$ indicates three identical policies. Over a discrete space of risk levels we then find the $p_\rho$ that maximizes $\sigma$. A total of $5,000$ random states were sampled where the average $\sigma$ was used to select the final $p_\rho$. This procedure was conducted for each layout.

As seen in Fig. 6, $p_\rho = 0.4$ was selected for RCR and $p_\rho = 0.15$ was selected for RMP. Several $p_\rho$'s achieved comparable score leading the possibility of alternative risk levels being viable for simulation. However, for conciseness, we strictly selected the maximum value.

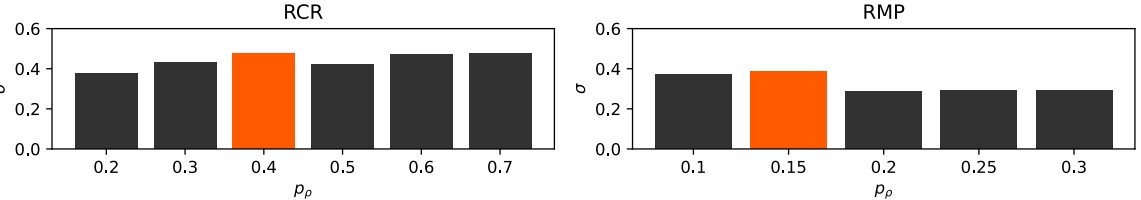

*Figure 6.* Mutual dissimilarity scores $\sigma$ for different risk-levels $p_\rho$ in the RCR (left) and RMP (right) layouts. Orange bars indicate the selected risk-level that maximizes $\sigma$.

### B.3. Timeseries Snapshots

To supplement the cumulative expression of joint-policies in Fig 3, this section will more directly examine how strategies differ with risk-sensitivity using illustrative examples. In Fig. 7, we see four brief timeseries snapshots of these strategies. In all snapshots, agents are attempting to either bring onions or dishes to the pot in order to complete subtasks 1) or 2), respectively, as mentioned in Sec. 3.1.

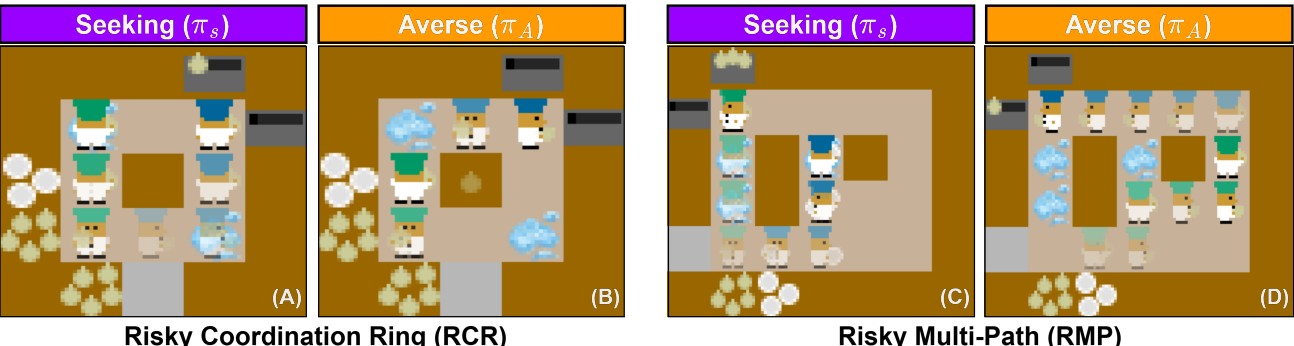

*Figure 7.* Illustration of timeseries snapshots of the risk-seeking $\pi_S$ and risk-averse $\pi_A$ strategies in the RCR and RMP layouts. More transparent chefs and objects indicate timesteps further in the past.

In (A), we see that both agents are optimistically traversing puddle states with onions in order to avoid possible coordination costs involved with handing off onions through the center counter tile. In contrast, (B) has the opposite preferences where pessimistic tendencies cause agents to view the consequences of slipping in a puddle to outweigh these coordination costs. Therefore, agents will form joint strategy that passes onions through the center counter tile.

In (C), an optimistic team will prefer to incur more risk by traversing puddle states in order to take a more direct route and avoid the added navigation costs of the long, risk-free detour. Conversely, in (D), agents prefer to incur these navigation costs and take the long detour as they pessimistically perceive the cost of slipping to be to great.

