# OpenReview forum: "Risk-Sensitive Theory of Mind: Coordinating with Agents of Unknown Bias using Cumulative Prospect Theory"
_ICML.cc/2025/Conference — ICML 2025 poster_

### Official Review · Reviewer_ah7K · 2025-03-10

**Overall Recommendation:** 4

**Summary:**

This paper presents a new method for learning agent strategies that support human decision-making under risk. The agent has the ability to learn strategies in situations where it does not know how risk-averse its partner is. The paper adopts prospect theory as a model for human decision-making and proposes a framework for determining strategies using reinforcement learning.

**Claims And Evidence:**

The research is based on the assumptions presented in previous literature, methods that have been proven effective in the field of agents, and evaluation experiments conducted in previous studies. As a whole, it is very convincing.

**Essential References Not Discussed:**

I believe the papers essential to this paper are included in the references.

**Experimental Designs Or Analyses:**

(Kwon et al., 2020; Sun et al., 2019; Danis et al., 2023; Ferreira et al., 2021)
What is the reason why these methods cannot be applied?
Even if they cannot be applied straightforwardly, I would also like to see the results of comparing them with methods that have been simply customized from these conventional methods.

**Methods And Evaluation Criteria:**

It is a reasonable fusion of human decision-making theory (prospect theory) and reinforcement learning. The evaluation experiment in this paper is based on the well-known setting (the Overcooked environment) by Carroll et al., and the game is extended in a way that involves risk in decision-making. I think it is appropriate as the most primitive simulation experiment to analyze decision-making under risk. On the other hand, as the author mentions in the Limitations, it has only been verified in a few simple scenarios. In addition, it is limited to simulation testing, and it has not been verified whether it is useful in actual decision-making scenes.

**Other Comments Or Suggestions:**

I don't understand why the methods of (Kwon et al., 2020 Sun et al., 2019 Danis et al., 2023 and Ferreira et al., 2021) cannot be applied to the current experimental setting, so I would like an explanation.

**Other Strengths And Weaknesses:**

- The structure of the paper is excellent. It was easy to understand the content.
- I can appreciate the interdisciplinary approach of economics and computer science.

**Questions For Authors:**

I have questions mainly about experimental methods and comparative methods.
See "Other Comments Or Suggestions" and "Experimental Designs Or Analyses".

**Relation To Broader Scientific Literature:**

In this research, behavioral economics (prospect theory) is used to model the human decision-making process. In other words, the basic behavior of agents and partners is based on these findings. This research can be positioned as interdisciplinary research in the fields of economics, the social sciences, artificial intelligence, and computer science related to human behavior.

**Theoretical Claims:**

The main proof is included in the reference materials from page 11, but I have not verified the accuracy of the proof.

---

> ### Author Rebuttal · Authors · 2025-04-01
>
> Thank you for your comments and feedback. We have addressed them below.
>
> > **C1:** The reviewer asked “(Kwon et al., 2020; Sun et al., 2019; Danis et al., 2023; Ferreira et al., 2021) What is the reason why these methods cannot be applied? Even if they cannot be applied straightforwardly, I would also like to see the results of comparing them with methods that have been simply customized from these conventional methods.”
>
> There are two main reasons discussed at the beginning of Sec. 3 but we have provided some additional elaboration below:
>
> 1. (Kwon et al., 2020; Sun et al., 2019) generate diverse risk-sensitive behaviors by relying on data-priors collected from a human partner. In a simulation study, we simply do not have this option but, more importantly, a key contribution of our algorithm is that we do not need a data prior to generating adaptation to risk-preferences of different human partners, because we have access to a cognitively valid decision model (i.e., CPT).
>
> 2. (Danis et al., 2023; Ferreira et al., 2021) rely on tabular methods like Q-learning algorithms which simply do not scale to large state-action spaces. We actually tried this and it did not go well because the Q-table must allocate memory for every state-joint-action pair. Other works have used low-level motion planners to simplify this problem and solve over high-level sub-tasks but this is not available to us due to “risk-sensitivity” of an agent being present in their low-level navigation policy (see Appendix A.2 for details). Let's look at the smaller of the two layouts (RCR) shown in our paper as an a motivating example:
>     - 2 players have 5 actions each for a total of 5^2 = 25 actions
>     - 2 players can each have 6 x and 6 y coordinates, 4 orientations, and 4 holding states {empty, onion, dish, soup} leading to a total of (6\*6\*4\*4)^2 = 331,776 combinations.
>     - Each of the 7 reachable counter spaces have 4 possible states {empty, has onion, has dish, has soup} for a total of 4^7=16,384 combinations
>     - Each of the 2 pots has 5 variations {nothing, 1 onion, 2 onion, 3 onion, finished soup} for a total of 5^2=25 combinations
>     - This brings us to a 25 \* 331,776 \* 16,384 \* 25 = 3.39e12 element array.
>     - With a 16 bit float precision, this requires 54 terabytes for each of the 3 models
>
>     As you can see, this quickly becomes infeasible to manage which requires a function approximation method when using this lossless state. We spent significant efforts to develop this novel deep learning algorithm including development of the novel replay memory which is one of our contributions to the AI community.
>
> We plan on adding “there is a lack of feasible baseline algorithms …[because] state of the art MARSRL algorithms (Danis et al., 2023; Ferreira et al., 2021) are not tractable in complex tasks like Risky Overcooked due to their tabular formulation creating infeasible memory requirements under the high number of permutations in the state space.” into Sec. 3 to remove this ambiguity in the revised paper.
>
> > **C2:** “it is limited to simulation testing, and it has not been verified whether it is useful in actual decision-making scenes.”
>
> Reviewer 3GLy had a similar comment which we recommend you check out the response to C1 for a full description of the summary we provide here. Here, there is a slight clarification about the claim of this paper that we intend to make more clear when listing the contributions.
>
> While testing “...whether it is useful in actual decision-making scenes” is the end goal of this research, that is not exactly the claim we are making in this particular paper. The claim we are making is that *we can overcome the challenges in state-of-the-art methods that already show adaptation to human risk-preferences improves human-robot/AI interaction*. Specifically, our approach overcomes:
>
> 1. the limited task complexity afforded by tabular RL methods (Danis et al., 2023; Kwon et al., 2020) through our “Deep MARSRL with CPT” algorithm.
> 2. the requirement for data priors to enable personalization (Kwon et al., 2020) through our generation of risk-sensitive candidate policies offline.
>
> We believe the simulations support this claim as we are able to generate diverse risk-sensitive behaviors in more complex coordination tasks (i.e. infeasible to solve with tabular methods) offline and correctly adapt to an agent of unknown risk-sensitivity online. Consequently, the algorithmic contributions are independently significant from human studies, fill an important gap in the existing literature, and fits the scope of ICML well. We are conducting these human experiments in an ongoing study that warrants its own paper. This way, we can maintain the focus on AI in the current ICML paper and shift towards a more human-centered perspective in the next work.

---

### Official Review · Reviewer_3GLy · 2025-03-13

**Overall Recommendation:** 1

**Summary:**

This paper proposes a multi-agent RL method where agents model the other agents under the cumulative prospect theory (CPT) model. This allows better coordination with risk-averse or risk-seeking agents (an agent may have both tendencies depending on the context), which includes human agents.

The proposed method consists of three components: first a CPT-transformation is applied on the state-action values and transition probabilities. Second, Nash equilibria under these transformed values are found. Finally, Bayesian inference is performed to identify which equilibrium the other agent is going for.

**Claims And Evidence:**

It is not clear to me if this algorithm easily extends to n>2 agents. In the current setup, one of the agents is modeled as the biased agent (whose behavior can be modeled via CPT) and the other agent adapts to it. This makes sense for human-robot collaboration scenarios. However, what if there are multiple humans? In such cases, different humans may go for different equilibria, and it is not clear if the robot(s) will be able to decide how to coordinate with them. So maybe claiming this is a multi-agent algorithm is a little misleading, and it should be framed as a human-robot collaboration algorithm where there is only one human and one AI agent.

Similarly, the problem formulation implicitly assumes the agents share the reward function, meaning this is a completely collaborative scenario. While I am fine with this assumption, I think this should be noted earlier, because multi-agent systems would normally include competitive or mixed-sum settings, too.

While the main claim is to better model humans' decisions under CPT-related biases which will then lead to better collaboration, there is no evidence about it in the paper due to the lack of human experiments. The simulation experiments with artificial agents do not provide enough evidence, because they are controlled experiments that are already expected to work ("designed to work" in a sense).

**Essential References Not Discussed:**

N/A

**Experimental Designs Or Analyses:**

While I like the paper a lot in general, the lack of human experiments prevents me from giving an accept score. Because the target biases are human specific and it is impossible to evaluate the value of the paper without having human experiments.

**Methods And Evaluation Criteria:**

The proposed method makes sense but some assumptions should be stated more explicitly.

- One of them is the assumption about n=2 agents as I mentioned before.
- Another one is the shared rewards assumption, again as I mentioned before.
- It seems to me that approximating the Nash equilibria and the procedure described after line 165 (right column) require access to the transition model, i.e., access to the probability distribution over the next state given the current state and the action is needed. This should be stated earlier.

Also, while the paper starts defining an MDP, it includes multiple agents that are decentralized, so it should really be a multi-agent MDP.

**Other Comments Or Suggestions:**

There are several typos in the paper. Some of them are:

1) "a RS-ToM" --> "an RS-ToM"
2) "in in a zero-shot" --> "in a zero-shot"
3) "in this a deep multi-agent setting" --> "in this deep multi-agent setting"
4) "Figure. 2" --> "Figure 2"

**Other Strengths And Weaknesses:**

N/A

**Questions For Authors:**

N/A

**Relation To Broader Scientific Literature:**

While talking about the prior work, the paper states some of them "train policies under the median CPT parameter estimates from (Tversky & Kahneman, 1992) which generally describe risk-averse behaviors. I am skeptical about this statement. As far as I know (but I am happy to be convinced otherwise), the median parameters model both the risk-averse and risk-seeking behaviors depending on outcomes being negative (e.g., people buy insurance to avoid large losses) or positive (e.g. people buy lottery tickets to earn lots of money).

**Theoretical Claims:**

There is no theoretical claim in the main paper -- some "somewhat theoretical findings" are in the Appendix which I didn't review.

---

> ### Author Rebuttal · Authors · 2025-04-01
>
> We thank the reviewer for their feedback.
>
> > **C1:** The reviewer is concerned that “the claim is to better model humans' decisions under CPT-related biases which will then lead to better collaboration, there is no evidence about it in the paper due to the lack of human experiments”
>
> We would like to point out that there is “similar transfer between simulation and human studies” (Smith & Zhang, 2025) showing a side-by-side comparison between simulation and human studies to support external validity.  (Kwon et al., 2020) shows that modeling risk-sensitivity with CPT leads to improved performance and robotic perceptions. Thus, we see that there is existing support for modeling human decisions with CPT leading to better collaboration in simple settings but this is not directly our claim.
>
> The claim we are making in this paper is that *we can overcome AI challenges in state-of-the-art methods that already show adaptation to human risk-preferences improves interaction*. Specifically, we are addressing:
>
> 1. the limited task complexity available with tabular RL methods (Danis et al., 2023; Kwon et al., 2020) by using our “Deep MARSRL with CPT” algorithm affords scaling to scenarios more aligned with real world resolutions
> 2. the requirement for data priors to enable personalization (Kwon et al., 2020) using our generation of risk-sensitive candidate policies offline to resolve costly data collection when human risk is present.
>
> The simulations show the ability to generate diverse risk-sensitive behaviors in more complex coordination tasks (i.e. infeasible to solve with tabular methods) offline and correctly adapt to an agent of unknown risk-sensitivity online which supports this claim. Consequently, we believe the algorithmic contributions are independently significant from human studies, fill an important gap in the existing literature, and fit the scope of ICML well .
>
> We agree on the importance of supporting the “better model humans' decisions” claim that you mentioned. This paper paves the way for subsequent human studies. We have already started human experiments where we intend to collect 60 subjects where we employ more sophisticated experimental designs to characterize human behavior in response to RS-ToM and Rational agents. We believe that this next study warrants its own paper due to the possibility that combining these works would cause muddled contributions and decrease the amount of depth we can explore in each due to the page limit constraints.
>
> > **C2:** The reviewer is not clear how this algorithm extends to n>2 agents and believes it should be framed as a human-robot collaboration instead of a multi-agent problem.
>
> This work is motivated by the human-robot interaction paradigm. However, the MARSRL algorithm could be applied to generate risk-sensitive multi-robot interactions if desired. Additionally, the RS-ToM framework can work for n>2 agents by factorizing the state and action (e.g., $a_1$ x $a_2$ x…x $a_n$) in Algorithm 1 and extending the equilibrium solution (QRE with level-k reasoning) to initialize a level-1 player (see Appendix A.5) to assume all other agents have random policies where the remaining procedure remains the same. We will add an explanation of how to extend to n>2 agents into the end of Sec. 2.2.
>
> You also make an important point about how to align with humans “with different equilibria”. This is an ongoing and active research field called multi-value alignment where more sophisticated multi-objective optimization methods, such as the MAP algorithm (Wang et al., 2024), must be applied. We will discuss this at the end of Sec 2.3.
>
> >**C3:** The reviewer suggested stating the assumptions about (a) requiring access to the transition model and (b) a completely collaborative scenario earlier.
>
> We will add (a) to the beginning of Sec. 2 and (b) to the beginning of Sec. 3.1.
>
> > **C4:** The reviewer mentions that Def. 2.4 should be a multi-agent MDP.
>
> We agree and will state that Def. 2.4 defines a multi-agent MDP when the joint action and state spaces are factored out.
>
> > **C5:** The reviewer is skeptical about our statement “median CPT parameters …generally describe risk-averse behavior”
>
> You are correct in that both averse/seeking behaviors emerge from median parameters depending on context. We say “generally” to refer to the fact that, under random prospects, we expect the loss to be disproportionately represented in the choice since $\ell = 2.25$. Therefore, these parameters generate risk-averse preferences in most cases. We refer you to Appendix B.1 for additional discussion.
>
> > **References**
>
> Li, C., Wang, T., Wu, C., Zhao, Q., Yang, J., and Zhang, C. Celebrating diversity in shared multi-agent reinforcement learning. In Advances in Neural Information Processing Systems, volume 34, pp. 3991–4002. 2021.
>
> Wang, X., Le, Q., Ahmed, A., Diao, E., Zhou, Y., Baracaldo, N., Ding, J., and Anwar, A. Map: Multi-human-value alignment palette, arXiv preprint arXiv:2410.19198, 2024.

---

> > ### Comment · Reviewer_3GLy · 2025-04-03
> >
> > I thank the authors for their responses. I am happy to see they will update the paper based on the comments I made earlier. However, I still see the lack of human experiments as a critical issue. I understand that the previous papers already showed CPT model better explains human decisions, but as the paper itself acknowledges, those were limited to simple, tabular settings. Showing incorporating CPT models increases performance in more complex settings via simulations, without first showing humans still take their decisions following the CPT model is questionable. If the latter is not verified/correct, that would mean the paper is just solving a problem that does not exist in reality.
> >
> > So I still hold my belief that the paper must report human subject study results. It seems I am not the only reviewer who raised this issue (3 of the 4 reviewers did it), which I hope, further highlights the importance of such studies for this paper.

---

> > > ### Author Response · Authors · 2025-04-04
> > >
> > > We thank the reviewer for their response but respectfully disagree that this paper requires human studies for publication in ICML. We believe the critique that prevents the reviewer from recommending acceptance is that *we need human studies to validate that our approach has value in real world settings*. We have responded to this below:
> > >
> > > # 1. Support of Prior Work:
> > > We disagree that extending tabular to deep learning invalidates support from prior findings showing CPT can model humans. In regards to “the paper itself acknowledges, those were limited to simple, tabular settings,” we are pointing out that these approaches have practical limitations when learning policies in complex settings, not that the approach is inherently invalid if learning were possible.
> > >
> > > > **Example:** neglecting computational issues of tabular methods finding optimal policies, it is reasonable to expect that previous findings continue to apply as complexity is scaled up. Since deep learning is a function approximator of this optimal tabular policy, it is follows that we would recover similar findings given our approximation is good.
> > >
> > > Thus, we strongly disagree that this paper is possibly “solving a problem that does not exist in reality” as it is well motivated by prior works such as (Kwon et al., 2020; Ferreira et al., 2021).
> > >
> > > Empirical validation of CPT in *explicitly* complex settings simply do not exist due to the gap in algorithms that enable learning (i.e., our contribution). [(Gao et al., 2022)](https://doi.org/10.1016/j.ijdrr.2022.102904) mentions a similar challenge: “studies on applying CPT to transport-related problems have been limited to examining the choice of strategies or alternatives, and no study has used CPT to drive the movements [with finer resolution].” They do not address complexity at our scale but do show that their CPT model with finer resolution actions matched *real human data*. This implies CPT holds when extending to higher resolution policy spaces and addresses the reviewer’s concern if “humans still take their decisions following the CPT model” in more complex settings.
> > >
> > > # 2. Must Contain Human Experiments:
> > > We disagree that this work must report human subject study results to have sufficient value for ICML.
> > >
> > > **2.1 AI Contributions:** We are planning to conduct human experiments but, in order to do that, we must *first develop the fundamental AI approaches that enable future human studies* (i.e., without first having a model, we cannot validate it in human studies). During development of RS-ToM, we were able to make the aforementioned algorithmic contributions that are independently valuable to the AI community. From the reviewer’s initial comments, it seems they support these contributions up until the need for human studies. However, we show sufficient evidence in simulation that we can overcome the mentioned AI challenges and strongly believe that human studies warrant a separate paper for the reasons mentioned in the original rebuttal.
> > >
> > > **2.2 Simulations are a Common Paradigm:** Using simulations to validate human(-like) modeling is a common paradigm in this field. We have provided 3 works published in reputable AI conferences that follow our paradigm:
> > >
> > > 1. The ICML paper in [(Prashanth et al., 2016)](https://proceedings.mlr.press/v48/la16.html) asserts that “CPT realistically captures the attitude of the road users towards delays” and shows that their CPT algorithm better optimizes simulated humans’ subjective utility than “traditional expected delay optimizing algorithms”
> > >
> > > 2. The AAAI paper in [(Tian et al., 2021)](https://doi.org/10.1609/aaai.v35i7.16750) shows the model’s ability to generate CPT behaviors relative to risk-neutral baseline in simulation. They remark that “our solution provides an interpretable and heterogeneous human behavioral model.”
> > >
> > > 3. The AAMAS paper in [(Ghaemi et al., 2024)](https://dl.acm.org/doi/10.5555/3635637.3663134) shows generation of diverse CPT behaviors in simulation and connect to human modeling by remarking “…[is] a suitable framework to show the tangible effect of loss aversion in human-like agents.” and “A potential application of the proposed framework is calculating CPT risk-sensitive policies of human agents in real-world settings”
> > >
> > > **2.3 Other Reviewers:** We agree with all 3 reviewers that human experiments are an important step and in deployment of this work in realistic settings. However, we do not believe other reviewers agree that the lack of human studies warrants an insufficient contribution to ICML or that this research problem is unsupported by the current literature. While we acknowledge that there may be different opinions for scoping this work, this is an important and well-motivated step towards human modeling in complex settings with significant contribution to ICML.
> > >
> > > > **Final Remark:** We again thank the reviewer for their comment and hope that this evidence justifies the motivation and current scope of this paper

---

### Official Review · Reviewer_pniM · 2025-03-14

**Overall Recommendation:** 4

**Summary:**

This paper proposes a new model of risk-sensitive multi-agent coordination towards better aligning autonomous agents with human utilities. The authors define a risk-sensitive ToM that affords adaptation to a partner with unknown risk-sensitivity in in a zero-shot
fashion by pre-training several policies with different risk preferences, and online matching observed behavior to these policies.

**Claims And Evidence:**

Claim: The proposed model can collaborate with humans better than an AI partner without RS-ToM
Supported, within the scope of this task.

Might not generalize to beyond toy scenarios, in general the same behavior could arise not only from risk sensitivity, but also from other approximations to sequential decision-making (see suggested refs).

**Essential References Not Discussed:**

Overcooked as a model of Cooperation with ToM:
https://onlinelibrary.wiley.com/doi/full/10.1111/tops.12525

This paper evaluates prospect theory in human sequential decision making, alongside limited planning horizon, and a model of numerosity in distance/area perception, showing that prospect theory may indeed generalize to spatial planning, yet is not an exclusive cause of such deviations.
https://journals.plos.org/ploscompbiol/article?id=10.1371/journal.pcbi.1012582

**Experimental Designs Or Analyses:**

Yes

**Methods And Evaluation Criteria:**

Yes. I am very well familiar with Overcooked domain for Cooperation.

**Other Comments Or Suggestions:**

Edit: I have increased my rating by a point, to Accept.

**Other Strengths And Weaknesses:**

The paper is well written and original.

I'm leaning to accept, but I'd like to see a better discussion of limitations and related work.

**Questions For Authors:**

N/A

**Relation To Broader Scientific Literature:**

The lit review is mostly good, but a few references are missing.

1. The authors should discuss the advantages and the reasons for choosing a RL approach over Bayesian ToM  previously used with Overcooked. The authors claim to achieve zero-shot understanding of the human partner without prior collection of human data, however the downside to this method is the need for excessive per-training. How would this scale?

2. The authors take assumption of noisy-rationality as a baseline, however in field of modelling human planning, it is well established that people may be using a variety of approximations that are more elaborate than injecting noise, most notably limited planning horizon, heuristics, simplified space representations.
In the original BToM papers (e.g. Baker et al 2017) high-temperature noise is simply a convenient catch-all for non-alignment with the optimal policy, given that they use trivial scenarios such that will not allow to differentiate between the various approximations. I do not suggest reviewing this entire field of human planning models, but I suggest carefully framing the limitations.

**Theoretical Claims:**

Skimmed the math, seems fine.

---

> ### Author Rebuttal · Authors · 2025-04-01
>
> Thank you for your comments.
>
> > **C1:** The reviewer commented on the generalization of our algorithm beyond these toy scenarios and that alternative models may give rise to the same behavior. They suggest that we mention this in limitations.
>
> We generally agree that the observed behaviors could arise from alternative models (e.g., limited planning horizon). However,
>
> 1. it is unlikely that alternative models would consistently differentiate risk-averse/seeking policies and that the novelty of our approach has distinguishable value across use-cases and in more complex tasks.
> 2. As for generalizing beyond toy scenarios, we chose to extend Overcooked for this very reason: it is a well accepted baseline for coordination tasks that is simple enough for validation but rich enough to allow diverse coordination strategies. Once validated, we can scale to more complex domains where higher differentiability is available. In the second paragraph of Limitations and Future Work, we mentioned “increasing the resolution of the candidate policies may be helpful for scaling to tasks with a more diverse set of risky-decisions available (e.g., continuous control problems affording continuous variation in risk-sensitive strategies)”.
>
> To address these two components, we add the following discussion to the Limitations section:
>
> - “Due to the limited number of strategies (e.g., traverse puddle or detour), alternative models, like limited planning horizon and heuristics, could potentially give rise to the observed behaviors. However, these similarities are likely to fade as decisions become more complex and afford more variation in behaviors. While we believe that modeling risk-sensitivity is an essential factor in risky settings, humans are subject to many biases and heuristics. Therefore, a multimodal approach (Kryven et al. 2024) that captures risk-sensitivity along with other human planning characteristics will be pursued in future work.”
>
> > **C2:** The reviewer suggested discussing why we took our RL approach instead of the “Bayesian ToM previously used with Overcooked.”
>
> As you pointed out, the Bayesian ToM approach was used in (Wu et al. 2021) where they formulate their ToM as reasoning over intentions and sub-tasks. However, we reason over risk-sensitive preferences in more complex tasks and our RL-approach is more generalizable since it
>
> 1. Defines continuous variation over a larger latent space (i.e. CPT parameters)
> 2. Does not need manual re-definition of mental states (i.e. sub-tasks) when working on a new task
> 3. Applies deep RL to overcome state space complexity issues that exist in their tabular bounded real-time dynamic programming (BRTDP) approach to policy solutions.
>
> We will add discussion of this article when introducing ToM and Overcooked.
>
> > **C3:** The reviewer mentioned that the focus on pre-training is a downside and asked "How would this scale?”
>
> We view the focus on pre-training as an advantage where we address excessive pre-training and scaling in the following ways:
> 1. Pre-training is more affordable/scalable than current approaches that require collection of human data priors for personalization (Kwon et al., 2020), especially when risk to the human is present. Additionally, we can aggregate an increasing resolution of an RS-ToM over time which affords arbitrary complexity that meets the specific needs of the more sophisticated tasks.
> 2. We do not need to continue training to adapt to more humans. Once we have completed the pre-training, we have a fully deployable model for all types of human partners. It is very scalable in this sense.
> 3. The future work section includes an approach to reducing pre-training requirements by using model interpolation methods but we will also add discussion of transfer learning (Weiss, 2016) to better address the challenge of scaling.
> 4. This algorithm can also scale to teams with more than two agents. See Reviewer 3GLy’s response to C2 for details.
>
> > **C4:** The reviewer suggested references of (Wu et al. 2021) and (Kryven et al. 2024).
>
> We have incorporated discussion of these papers in the sections mentioned in the previous response.
>
> > **C5:** “Did not see SI”
>
> We included the SI in the same PDF as the main paper.
>
>
> > **References**
>
> Houlihan S. D., Kleiman-Weiner M., Hewitt L. B., Tenenbaum J. B. and Saxe R. Emotion prediction as computation over a generative theory of mind, Philosophical Transactions of The Royal Society. 2023.
>
> Kryven, M., Yu, S., Kleiman-Weiner, M., Ullman, T., and Tenenbaum, J. Approximate planning in spatial search. PLOS Computational Biology, 20(11):1–21, 11 2024.
>
> Weiss, K., Khoshgoftaar, T.M. & Wang, D. A survey of transfer learning. Journal of Big Data, 3, 9. 2016.
>
> Wu, S. A., Wang, R. E., Evans, J. A., Tenenbaum, J. B., Parkes, D. C., and Kleiman-Weiner, M. Too many cooks: Bayesian inference for coordinating multi-agent collaboration. Topics in Cognitive Science, 13(2):414–432, 2021.

---

### Official Review · Reviewer_2s5A · 2025-03-14

**Overall Recommendation:** 3

**Summary:**

The authors consider the problem of a two-agent cooperative sequential decision task, in which one of the agents is a human, and address the problem of learning to coordinate behavior in the second agent. The idea is to model human behavior as following cumulative prospect theory, i.e. scaling probabilities and values according to a specific family of functions. By inferring which such function from a set of predefined parameters most closely describes human behavior, the agent can leverage this model to solve the multi-agent choice problem using DDQN. This involves computing the expected action of the human agent with level-k reasoning. Transitions are stored in a buffer for computational efficiency. The algorithm is applied to synthetic data with different simulated human agents in a new variant of the overcooked task and it is shown that performance in coordination is close to optimal as quantified by an oracle baseline and significantly improved with respect to a model of the other agent assuming noisy optimality.

**Claims And Evidence:**

The main motivation for the proposed model is that it captures human behavior better than alternative methods, as it models human choices with cumulative prospect theory. However, the evaluation is done exclusively on synthetic data. Human experiments would have been more convincing. For the synthetic data, the claims are convincing, both the inference of the agent's risk attitude as well as performance.

**Essential References Not Discussed:**

Maybe citing some work on inverse RL may be adequate, particularly IRL work that allowed the agent to be suboptimal with respect to the known rewards such as work involving soft policies or work connecting IRL to preference elicitation?

**Experimental Designs Or Analyses:**

I could not check any code or simulations but the evaluations in terms of risk attitude and performance seems sound.

**Methods And Evaluation Criteria:**

The newly devised risky overcooked task makes sense for evaluating cooperation with potentially nonneutral risk attitudes.

**Other Comments Or Suggestions:**

Typos:
“Humans like are said to have“
“they are holding where rationale for how values for pρ can be found “
“risk-sensitivity We also”

**Other Strengths And Weaknesses:**

Definition 2.3 could be easier to read by writing the bounds on the sums in the initial definitions in terms of l instead of two different ks, especially because the expectation then uses l.

**Questions For Authors:**

How does moving from quantal response equilibrium to an approximate action from the agent using level-k reasoning affects the proof of convergence?

Why did you add a decaying weight to the likelihood P(O|π) of observations further in the past?

Why did you not include at least a limited human study as the motivation is to better model human behavior?

**Relation To Broader Scientific Literature:**

The paper sits somewhere between multi-agent RL, cognitive science, inverse RL, and alignment in human-machine interaction.

**Theoretical Claims:**

The proofs involve standard contraction mapping arguments. Adapting to cumulative prospect theory, one can leverage the fact that the utility weighting functions are monotonically non-decreasing. However, I am not sure how moving from quantal response equilibrium to an approximate action from the agent using level-k reasoning affects the argument.

---

> ### Author Rebuttal · Authors · 2025-04-01
>
> We appreciate your effort in providing comments and respond to them below.
>
>
> > **C1:** The reviewer asked why we did not include a limited human study.
>
> We recommend you check the response to Reviewer 3GLy’s C1 for a full description of the summary we provide here.
>
> While “...capturing human behavior better than alternative methods” is the end goal of this research, that is not exactly the claim we are making in this particular paper. The claim we are making is that *we can overcome the challenges in state-of-the-art methods that already show adaptation to human risk-preferences improves human-robot/AI interaction*. Specifically, our approach overcomes:
>
> 1. the limited task complexity afforded by tabular RL methods (Danis et al., 2023; Kwon et al., 2020) through our “Deep MARSRL with CPT” algorithm.
> 2. the requirement for data priors to enable personalization (Kwon et al., 2020) through our generation of risk-sensitive candidate policies offline.
>
> We believe the simulations support this claim. Consequently, the algorithmic contributions are independently significant from human studies, fill an important gap in the existing literature, and fits the scope of ICML well. We are conducting these human experiments in an ongoing study that warrants its own paper. This way, we can maintain the focus on AI in the current ICML paper and shift towards a more human-centered perspective in the next work.
>
> > **C2:** The reviewer asked how the level-k reasoning affects the convergence proof.
>
> Please check out Proposition A.6 and the supporting Definition A.4 and Definition A.5. Here, we leverage the fact that this is a common-payoff game to show there exists a globally optimal point to every stage game (Assumption A.3) and that the quantal response equilibrium converges to this globally optimal game solution as $\lambda\rightarrow\infty$.
>
>
> > **C3:** “Maybe citing some work on inverse RL may be adequate, particularly IRL work that allowed the agent to be suboptimal with respect to the known rewards such as work involving soft policies or work connecting IRL to preference elicitation?”
>
> While our related work is focused on RL to support the contribution of our RL algorithm, we acknowledge the relevance of IRL and did previously include a CPT-IRL paper that related to this work (Sun et al., 2019). This was to contextualize an alternative to our approach and illustrate a major hurdle that our work seeks to address: personalized adaptation to risk-preferences (e.g., learned from IRL) requires a human data prior which may be costly or infeasible to collect, especially when human risk is present. This is one of the primary advantages of using a cognitively valid model of human behavior (i.e., CPT) in that we do not need a data prior to achieve human models in risky settings.
>
> To address this comment, we added review of additional work on IRL to better contextualize the problem:
>
> 1. (Rothkopf & Dimitrakakis, 2011) A more standard approach to Bayesian IRL for preference elicitation when observing suboptimal demonstrations.
> 2. (Cheng et al., 2023) Applies a more interactive approach risk-sensitive to IRL through interactive querying.
> 3. (Bergerson, 2021) The authors identify several approaches to multi-agent IRL that identify suboptimal human traits (i.e. noise, biases and heuristics) but the existing work shows limited attention on integrating cognitive models like prospect theory.
>
> > **C4:** The reviewer suggested notational edits to Definition 2.3 and pointed out a few typos.
>
> We have updated these items.
>
>
> > **C5:** The reviewer asked what the purpose of the decaying weight of previous observations in the belief update was for.
>
> This is intended to accommodate human agents shifting their risk preferences online where newer observations are more representative of their risk preferences at the current moment. Using a forgetting factor like this is a common approach in recursive inference to ensure stability and representativeness of the inferred parameter (Liu et al., 2016). While the simulated agents in this paper are fixed, the decay is intended to emulate settings with real humans to maintain external validity. We will add an explanation of this into Sec. 2.3.
>
>
> > **References**
>
> Bergerson, S. Multi-agent inverse reinforcement learning: Suboptimal demonstrations and alternative solution concepts. Computing Research Repository, abs/2109.01178, 2021.
>
> Cheng, Z., Coache, A., and Jaimungal, S. Eliciting risk aversion with inverse reinforcement learning via interactive questioning, arXiv preprint arXiv:2308.08427, 2023.
>
>
> Liu, C., Zhang, W., and Tomizuka, M. Who to blame? learning and control strategies with information asymmetry. In 2016 American Control Conference, pp. 4859–4864, 2016.
>
> Rothkopf, C. A. and Dimitrakakis, C. Preference elicitation and inverse reinforcement learning. In Machine Learning and Knowledge Discovery in Databases, pp. 34–48, 2011.

---

### Decision · Program_Chairs · 2025-05-01

**Decision:**

Accept (poster)

**Comment:**

This paper investigates a two-agent sequential decision-making problem, where one agent learns to coordinate with a human partner whose behavior is modeled using cumulative prospect theory. The authors propose a risk-sensitive reinforcement learning approach and demonstrate its effectiveness through empirical evaluation in simulated environments.

A key limitation of the work is the absence of experiments involving actual human subjects. Although the paper focuses on modeling human-specific biases, the presence and impact of such biases are not directly validated within the experimental settings presented.

Following extensive discussions with the reviewers, a consensus was not fully reached on this issue. Nonetheless, there is agreement that the paper offers substantial methodological and algorithmic contributions, and that the proposed approach is convincingly validated in simulation. It is strongly recommended that the limitation regarding the lack of validation with real human participants be clearly acknowledged in the camera-ready version.